# Recent Advances in Palladium Nanoparticles-Based Hydrogen Sensors for Leak Detection

**DOI:** 10.3390/s19204478

**Published:** 2019-10-16

**Authors:** Cynthia Cibaka Ndaya, Nicolas Javahiraly, Arnaud Brioude

**Affiliations:** 1Laboratoire des Multimatériaux et Interfaces, UMR 5615 CNRS-Univ Lyon 1, Université Claude Bernard Lyon 1, F-69622 Villeurbanne CEDEX, France; 2Laboratoire des Sciences de l’ingénieur, de l’informatique et de l’imagerie, ICube UMR 7357 CNRS- UniStra Equipe MaCÉPV, Université de Strasbourg, 23 rue du Loess, BP 20 CR, 67037 Strasbourg CEDEX 2, France

**Keywords:** gas sensors, Pd nanoparticles, hydrogen

## Abstract

Along with the development of hydrogen as a sustainable energy carrier, it is imperative to develop very rapid and sensitive hydrogen leaks sensors due to the highly explosive and flammable character of this gas. For this purpose, palladium-based materials are being widely investigated by research teams because of the high affinity between this metal and hydrogen. Furthermore, nanostructured palladium may provide improved sensing performances compared to the use of bulk palladium. This arises from a higher effective surface available for interaction of palladium with the hydrogen gas molecules. Several works taking advantage of palladium nanostructures properties for hydrogen sensing applications have been published. This paper reviews the recent advances reported in the literature in this scope. The electrical and optical detection techniques, most common ones, are investigated and less common techniques such as gasochromic and surface wave acoustic sensors are also addressed. Here, the sensor performances are mostly evaluated by considering their response time and limit of detection.

## 1. Introduction

Hydrogen is without doubt known as a very promising energy carrier in the development of a sustainable worldwide economy, improving storage and distribution of energy [1,2]. It is for example used as a fuel in space applications [3] and automobile industry [4]. As a matter of fact, since the Conference of Parties 21 (COP 21) that was held in Paris in 2015 to discuss climate change issues, the use of hydrogen as a clean energy carrier has been more and more promoted. In this scope, for example, in Paris (France), a partnership named Hysteco was recently signed between four companies in order to develop a network of 600 hydrogen taxis that would emit no carbon dioxide and only water. The agreement between Toyota, Air Liquide, Idex and Société du Taxi Electrique Parisien targets to put in service these hydrogen vehicles in 2020. Besides this, Air Liquide also recently announced the construction of the world largest hydrogen electrolysis plant in Bécancour (Canada) in order to meet the growing demand for carbon-free hydrogen in Canada and the United States. These examples illustrate that the use of hydrogen as a clean energy vector is really a topical issue.

The use of hydrogen, however, also means the manipulation of a highly explosive and flammable gas (flammability limits of hydrogen in air from 4 to 75 vol%) with a low minimum ignition energy (0.017 mJ), high heat of combustion (142 kJ/g H_2_) and a high burning velocity as well as an ignition temperature of 560 °C [3]. In addition, H_2_ being the smallest and lightest molecule, it has a high permeability through many materials [3,5]. It is then imperative to detect rapidly and accurately hydrogen leaks in order to prevent the risk of an explosion. Hence the need to develop researches on new ultrasensitive and ultrafast nano-sensors for the detection of hydrogen leaks is of main importance for security reasons.

As reported by Hübert et al. [3], several important parameters need to be taken into account in the development of hydrogen sensors including the response time, the detection range, signal accuracy, chemical selectivity, recovery time, low cost, low power consumption and low sensitivity to environmental parameters (relative humidity, pressure, etc.). 

To determine the efficiency of the hydrogen leak detection by the device, the sensor response time, chemical sensitivity and detection range parameters are particularly relevant. The response time is generally defined as the needed time to reach 90% of the signal maximum from the introduction of hydrogen in the sensor environment [1,6,7,8,9,10,11,12]. Accompanying the response time, the recovery time is commonly defined as the time needed to lower the signal maximum of 90%. Talking about hydrogen leak detection, the lower concentration that a device can detect, also called limit of detection (sometimes found noted as LOD) is an important parameter since to prevent explosion risk, the lower the detected concentration is, the better it is. In the present document, we will take these parameters (response time and limit of detection) as reference to evaluate the performances of the investigated hydrogen sensors. In the following, the considered limit of detection will be taken as an order of magnitude of the smallest hydrogen concentration inducing an observable device answer.

Depending on the application, the requirements on sensor performances could vary. As a matter of fact, the U.S. National Renewable Energy Laboratory (NREL) published a workshop report [13] presenting the sensor specifications expected in various domains where hydrogen is used. For example, in fuel cells automobiles, it would be appropriate for on-board safety sensors to be operational in the range of temperature within −40 °C to +40 °C and within 5% to 95% relative humidity. They should present a lower detection limit of at most 0.1 vol% of H_2_ with a response time smaller than 1 s at 1 vol% of H_2_ and a desired lifetime of 10 years without calibration or maintenance. In the case of hydrogen sensors deployed for safety of indoor hydrogen storage, the working temperature range is still estimated within −40 °C to +40 °C but this time from 15% to 100% relative humidity. The adequate limit of detection should be at most 0.4% with a response time lower than 30 s at 1 vol% H_2_ and a desired lifetime of 10 years. Although discrepancies are observed in the sensor performances specifications according to the targeted application, the Department Of Energy (DOE) in The United States provided a short list (given in Table 1) of target sensor performances to guide sensors developers in order to meet the needs of hydrogen community [13,14]. 

Numerous technologies have been developed for detecting hydrogen. Hydrogen detection principles often results from effects induced by the interaction of hydrogen with a selected sensing material. These effects can be catalytic-based, thermal conductivity-based, electrical and electrochemical-based, mechanical-based, optical-based as well as acoustic-based [3]. The use of several materials such as palladium-based, platinum-based, SiGe, metal oxides, etc. has been reported for hydrogen sensing applications [3]. Among them, the use of palladium and platinum materials has been and is still deeply investigated by several research teams due to their high sensitivity to hydrogen; especially Pd because of its very high hydrogen absorption capacity. Moreover, the use of Pd nanoparticles (NPs) may provide improved performances compared to the use of bulk Pd due to the increased surface to volume ratio that induces a higher effective surface available for interaction of Pd with the hydrogen gas molecules [15]. Thanks to the use of nanoparticles, it may be possible to engineer the sensor response time through materials design. The sensor performances could also be improved by tailoring the particles dimensions which would reduce gas diffusion times in the sensing Pd-based NPs [11].

In the present review, advances on hydrogen detection achieved using palladium NPs-based materials regarding their response times and limit of detection performances will be addressed. For a better understanding of the sensing mechanisms, a first section will be dedicated to the study of interactions between palladium and hydrogen.

## 2. Review of Palladium-Based Hydrogen Sensors

### 2.1. Pd-H_2_ Interactions

Palladium is well known for its high affinity with hydrogen which makes it a suitable material for hydrogen storage and sensing applications. It was the first metal in which hydrogen was reversibly introduced [16]. At room temperature, it can absorb 900 times its equivalent volume of H_2_ [17]. In this section, the interactions between these two species and particularly hydrogen in its gaseous form will be discussed.

When Pd samples are placed in a H_2_ gas enriched environment, absorption of hydrogen by the Pd structure is observed. Absorption of hydrogen by palladium is exothermic and at the equilibrium state follows the reversible Equation (1) [18,19,20]. This phenomenon starts with a displacement of H_2_ gas molecules towards the Pd surface where they interact with Pd atoms through Van der Waals forces. The potential energy of the gas molecules shows a minimum at a distance of approximately one molecular radius inducing the adsorption of H_2_ gas at the metal surface [19]. The adsorbed gas molecules are then dissociated into H atoms and they diffuse in the metal structure [18,19]. At room temperature, the hydrogen diffusion coefficient in palladium is at the order of 10^−7^ cm^2^·s^−1^ [18,21,22]. All the steps listed here are illustrated in Figure 1. 

(1)$$\mathrm{Pd}+\frac{x}{2}H_{2}↔{\mathrm{PdH}}_{x}$$

The diffused H atoms occupy interstitial sites with a preference for octahedral sites resulting in a partially filled NaCl type structure. Insertion of hydrogen in the Pd crystalline structure induces a structural reorganization of the host metal that leads to phase transformation of the Pd lattice network [18]. This latter phenomenon is defined by three parameters: equilibrium hydrogen pressure, temperature of the system and hydrogen concentration in Pd [18,19,20,23]. Therefore, Pressure- Composition-Temperature (P-C-T) diagrams can be established to investigate the hydrogen-induced changes in the metal structure. A P-C-T diagram is obtained by performing series of isothermal measurements where the hydrogen concentration in Pd is determined as a function of the hydrogen partial pressure in the medium at a set temperature. A Pd P-C-T diagram is presented in Figure 2a. In fact, Figure 2a represents an ideal P-C-T diagram and the differences with a real diagram will be discussed later. Three distinct parts could be distinguished in the Pd P-C-T diagram. The first part is the α branch: a solid solution (α phase) of hydrogen in the metal structure is formed when a relatively low hydrogen pressure is applied. 

The system is composed of two species: gaseous hydrogen and metallic Pd. Hydrogen concentration in the solid solution increases with an increase of H_2_ partial pressure until a saturation value α_max_ at a set temperature. At room temperature, α_max_ is about 0.02 [12]. The second part is the equilibrium plateau where α- and β-phases coexist: when α_max_ is reached, the increase of H_2_ partial pressure induces the formation of a new phase, the palladium hydride (β phase) according to the reversible Equation (2). The system is composed of three species: gaseous hydrogen, metallic Pd and Pd hydride. At a fixed temperature, this reaction takes place at a constant pressure until the complete transformation of α phase into β phase (Pd hydride with a composition β_min_) is achieved. This pressure is called the equilibrium dissociation pressure [18,19,20]. The increase of hydrogen gas amount in the system results in more Pd hydride formation while the pressure in the system remains constant. At room temperature, β_min_ is about 0.57 [23] (often taken as 0.6 [12]). The third part of the Pd P-C-T diagram is the β branch: the increase of H_2_ pressure induces increase of β phase formation with a higher hydrogen concentration in palladium hydride. However, a stoichiometric PdH is almost never achieved except in very high H_2_ pressure or very low temperature conditions due to limits imposed by electronic parameters [18,23]. Figure 2b illustrates the discussed phases.

(2)$${\mathrm{PdH}}_{\alpha_{\max}}+\frac{\beta_{\min}-\alpha_{\max}}{2}H_{2}↔{\mathrm{PdH}}_{\beta_{\min}}$$

The P-C-T diagram is used to describe the thermodynamic interactions between palladium and hydrogen. Indeed, as seen in Figure 2a, it gives access to the equilibrium pressure whose logarithm can be plotted vs 1/T in order to obtain the Van’t Hoff plot: a straight line whose slope is proportionnal to the change in enthalpy [19]. The equilibrium pressure is therefore related to the changes of enthalpy and entropy. The change of entropy is mostly associated to the change from molecular hydrogen gas to dissolved hydrogen. The change in enthalpy describes the stability of the metal–hydrogen bond [24]. 

As mentioned by Segard [18], one should note that when the temperature increases, the equilibrium dissociation pressure increases and the equilibrium plateau width decreases until reaching a critical state characterized by a critical temperature, pressure and composition respectively 563 K, 19 bars and 0.257 in the case of Pd bulk samples. Beyond the critical temperature, palladium and hydrogen are miscible in all proportions: a continuous solid solution of hydrogen in Pd is obtained.

In experimental conditions, the Pd P-C-T diagram is slightly different from an ideal one (Figure 2a). One major difference between these two diagrams, as can be observed in Figure 3, is the existence of a hysteresis phenomenon upon absorption and desorption of hydrogen [18,25,26]. Indeed, at a fixed temperature, the absorption pressure is higher than the desorption pressure. Moreover, α_max_ (absorption) is higher than α_max_ (desorption) and β_min_ (absorption) is higher than β_min_ (desorption). A hysteresis of about the logarithm of the ratio between the absorption and desorption pressures [18,27] is observed. The origin of the hysteresis phenomenon is related to the fact that phase transformations may not occur at thermodynamic equilibrium state due to high energy barrier to be overcome in order to favor the transformation [28,29,30]. The phase transformation generates stress in the material structure [29]. The hysteresis gap depends on the stress-induced change on the elastic properties of the materials and on the lattice mismatch between α and β phases [28,29]. In Pd bulk samples, plastic deformations occurring during α- to β- and β- to α-phase transformations are known to contribute to hysteresis due to their irreversible condition [18]. Hysteresis phenomenon could be at the origin of H_2_ sensor response inaccuracy, that could reach up to 45% of uncertainty, in the development of Pd-based H_2_ sensors [11]. It could also be at the origin of a lack of efficiency for Pd-based devices for hydrogen storage [18]. 

Furthermore, a non-constant equilibrium pressure in the coexistence zone of α and β phases is observed in an experimental Pd P-C-T diagram. Due to internal constraints during hydride formation, the equilibrium pressure increases slightly upon α to β phase transformation and a slope appears in the equilibrium plateau of the P-C-T diagram [18]. It should be underlined that reversibility is only partial in an experimental P-C-T because part of the absorbed hydrogen could remain deeply trapped in the metal host during desorption [18]. 

Because of hydrogen absorption in Pd structure, internal constraints are generated. They induce elastic (reversible) deformations in the α and β branches of the Pd P-C-T diagram while plastic (irreversible) deformations are induced in the coexistence zone of α- and β-phases. In the α-branch, an increase of the lattice parameter of about 0.1% is observed. At room temperature, it varies from 3.890 Å to 3.895 Å [18,31]. In the β branch also, an increase of the lattice parameter of about 0.1% is observed starting from 4.029 Å at room temperature for a β_min_ composition of the material [18,32]. In the α+β branch, an increase of the lattice parameter of about 3.5% is observed (from 3.895 to 4.029 Å at room temperature) [33]. However, as the temperature increases, the difference between lattice parameters values corresponding to α_max_ and β_min_ becomes smaller which is in accordance with the narrowing of the equilibrium plateau as the temperature increases [18]. The linear increase of lattice parameter upon Pd hydrogenation generates a linear and isotropic volume dilatation as a function of the material composition. Moreover, due to the described deformations in the crystalline structure, dislocations may be formed in the material [18,34]. 

As reported by several authors [35,36,37], occupation by H atoms in the interstitial sites of Pd crystalline structure induces electronic changes of the Pd sample. Indeed, due to hydrogen absorption, the width of the Pd valence d-band is diminished and new electronic states at energies just below the bottom of the band are induced. In addition, the increase of H concentration in PdH_x_ leads to a strong reduction of the density of states at the Fermi level for x ≥ 0.6 [35,36,37]. As a consequence of these electronic changes, electrical and optical properties of the material are modified. Several authors [38,39] have reported a decrease of work function and increase in electrical resistivity from Pd to PdH_x_ structures. Also, Silkin et al. [35], thanks to first principles calculations, reported the evolution of PdH_x_ dielectric function with hydrogen concentration in bulk Pd for x values ranging from 0 to 1. They found that upon increase of H concentration x in PdH_x_ which induces the change in PdH_x_ dielectric function, the plasmon energy of PdH_x_ is a decreasing function and is accompanied with a slight redshift. Moreover, they found that for spherical nanoparticles, in addition to the fact that the plasmon energy of PdH_x_ with x increasing from 0 to 1 follows the trend observed in bulk Pd (decrease), it is also lowered as compared to bulk [35]. The variation of electronic, electrical and optical properties from Pd to PdH_x_ formation allows the use of Pd-based materials as promising candidates for high performance H_2_ sensing applications.

It is important to highlight that the changes in Pd properties upon hydrogenation/dehydrogenation could be size dependent and therefore not similar when comparing the effects in a bulk or a nanoscale Pd sample. Several teams have investigated the thermodynamics of hydrogen interactions with Pd NPs [28,30,40,41,42]. While each of them provided key elements in the understanding of those interactions, Griessen et al. [27] recently managed to gather all this information in order to propose a general approach in the study of Pd-H_2_ thermodynamics interactions. They reported that unlike bulk Pd behavior which is consistent with incoherent hydrogen absorption and desorption processes, Pd NPs present a hybrid model of interaction with hydrogen where absorption is consistent with a coherent process while the hydrogen desorption process is mostly incoherent as in bulk. During a coherent phase transformation, the variations of spatial hydrogen concentration do not give rise to disruption in the Pd lattice as it is observed in Figure 4a,d [27] but to a smooth variation of lattice parameters between α and β phases. A single metastable state is generated instead of two distinct phases [27,28,29]. During an incoherent phase transformation, disruptions of the palladium lattice arise: lattice mismatches between the two distinct phases are observed [27,43]. Nucleation of one phase into the other takes place and coexistence of α-phase and β-phase occurs as it is seen in Figure 4b,c [27]. 

According to several works [28,41] and as summarized by Griessen et al. [27], upon hydrogenation of Pd NPs, it is suggested that H_2_ first saturates the subsurface sites giving rise to a core-shell loading scenario with α phase in the core and β phase in the shell coherently bound [27,28,41]. 

Both scenarios, coherent and incoherent phase transformations are accompanied with the opening of a hysteresis gap between absorption and desorption isothermal processes [27,28]. It is known that thermodynamic equilibrium of the phase transformation is satisfied when the chemical potentials of the surrounding H_2_ gas and hydrogen in the host metal are proportional. The fact that phase transformations may not occur at thermodynamic equilibrium state generates a hysteresis phenomenon. This results from the high energy barrier to be surmounted in order to favor the transformation [28,29,30]. On the one hand, coherency stresses generates a macroscopic elastic energy barrier which is proportional to the sample volume [27,28,29]. Therefore, upon coherent hydrogen absorption, the increase in the chemical potential of the interstitials site must first overcome the macroscopic energy barrier before allowing the system that is in a single metastable state to transform into the concentrated β-PdH_x_ phase [27,28,29,44]. On the other hand, in incoherent transformations, the phase transition does not take place at thermodynamic equilibrium since the large interfacial energy barrier between the two phases prevents the transformation [28]. Nonetheless, in incoherent processes, since dislocations are created to minimize elastic stresses and nucleation (and grow) of the second phase in the first occurs, the hysteresis might then be significantly diminished compared to coherent processes [27]. 

The asymmetry between H_2_ absorption and desorption processes in Pd NPs reported by Griessen et al. [27] illustrates that different mechanisms of interaction might be involved during these two transformations of the system. For example, according to several works, absorption pressures might be size dependent while desorption pressures are not [28,41]. Another example is in the work of Syrenova et al. [41] who mentioned that the absorption equilibrium pressure is the only responsible of the difference in hysteresis width when varying particle size while desorption equilibrium pressure does not influence this change. Indeed, the shrink of hysteresis width with decreasing particle size has been observed by many authors [28,30,40,41,42,45]. Other particle size-dependent features in the thermodynamics of palladium hydrogen interactions are addressed in the literature such as the narrowing of the equilibrium plateau when the particle size is reduced [30,42,45]. In order to better understand the NPs size influence on the thermodynamic interactions between Pd NPs and H_2_, as well as the difference in mechanisms involved in hydrogen absorption and desorption in the Pd NPs, more studied are needed.

As a result of all the point addressed in the previous paragraphs, it is therefore not surprising that the material properties induced by hydrogen interaction with nanoscale Pd samples are not the same as in the case of bulk. The next sections of the present document will focus on published works that take advantage of the materials properties change from Pd to PdH_x_ formation at the nanoscale to develop performant H_2_ sensors.

### 2.2. Pd NPs-Based H_2_ Sensors

H_2_ interactions with Pd-based materials induce structural changes in these latter that mainly lead to variation of their optical and electrical properties [3]. Therefore, building a hydrogen sensor from properties of Pd-based materials often involves the fabrication of either an electrical or optical device. In literature, the most commonly found Pd NPs-based sensors for H_2_ detection are electrical. They take advantage of the change in resistivity or work function of the system upon H_2_ exposure. The present section will first focus on electrical H_2_ sensor involving the use of Pd NPs. In the second part, optical devices will be investigated. Finally, less common systems for Pd NPs-helped H_2_ detection will be evoked.

#### 2.2.1. Electrical Sensors

Several teams worked on this type of sensors. Typically, an experiment is performed by measuring the system resistance or conductance change when switching from the carrier gas to H_2_ enriched environment in a cyclic manner inside a gas chamber in presence of a sensing sample [1,6,7,8,33]. During experiments, gas rates are monitored using mass flow controllers (MFC). A standard experiment setup, as used by Joshi et al. [7], is presented in Figure 5.

The sensor response is usually obtained from Equation (3) where RG is the initial resistance in the selected carrier gas and RH the resistance when exposed to H_2_ mixture [6,7,8,9,39,46,47,48] The answer, given as the relative variation of resistance from 0 to 100%, results from the ratio of the change in resistance when exposed to H_2_ enriched environment over initial resistance in the selected carrier gas. In the same way, some authors evaluate the response of their device by means of the relative variation of conductance [8,33] and some works also use the ratio of resistance [49,50,51,52,53]:(3)$$\mathrm{Sensor}\;\mathrm{response}\;\left(\%\right)=\frac{\left|R_{G}-R_{H}\right|}{R_{G}}\times 100$$

Electrical hydrogen sensors based on Pd NPs can be classified according to the configuration of the setup used. In these systems, Pd NPs can be found deposited directly on silicon or silica substrates [1,6,7,8,33]; they can also be deposited on other oxide materials (ZnO, WO_3_, etc.) [49,50,51,52,53,54,55,56,57,58,59,60,61,62,63,64,65]; in some works they functionalize carbon materials (graphene, CNTs, etc) [9,39,46,66,67,68,69,70,71] or are even alloyed to other metals [55]. These various types of sensors are investigated here.

##### Pd NPs on Si or SiO_2_ Substrates 

Here, as illustrated in Figure 6 [39], the gas sensing setup is generally composed of a Pd NPs layer directly deposited on top of a Si or SiO_2_ substrate. Metallic electrodes are added either on top or at the rear face of the system and connected with thin copper wires to a multimeter to measure the resistance or conductance change upon H_2_ exposure. Several teams have done remarkable experiments on this hydrogen sensor type investigating different Pd NPs synthesis and deposition technique (electrochemical deposition, spin coating, gas evaporation technique, etc.), different particle sizes and density, different positioning of electrodes as well as different signal recording systems [1,6,7,8,33]. Villanueva et al. have worked on an alternative configuration that consists on electrodeposition of an overgrown discontinuous array of Pd NPs inside the SiO_2_ layer and achieved to obtain good sensor response from 0.1% up to pure H_2_ at room temperature [1]. 

The sensors depicted in this section usually work at room temperature [1,6,7,8,33], however some researchers have developed devices that show good performance up to 50 °C [6]. The detection range of the present systems commonly lies in the low H_2_ concentration range, typically from 0.1% to less than 5% which makes them good candidates for leak detection [6,7,8,33]. It is known that the lower explosive limit of hydrogen concentration in air is 4%. 

Most of the experiments on the present devices are performed using N_2_ as carrier gas [1,6,7,8,33]. For comparison, Gupta et al. performed measurement in H_2_/N_2_ and H_2_/air mixtures [6]. They obtained less good results when selecting air as the carrier gas, while in N_2_, a response time of 3 s is obtained for 0.1% of H2 at 50 °C, a response time of 33 s is obtained in the same conditions when performing the experiment in air. Moreover, the maximum sensor answer obtained in N_2_ was 15.4% while in air it only reached 10.2%. Table 2 from [6] illustrates this comparison. 

The evolution of the sensor response with H_2_ concentration presented in Table 2 [6] describes the commonly trend obtained with this kind of devices in low range H_2_ concentration. Sensor response increases with H_2_ concentration. It is worth noting that the typical response time with this type of sensors in low range H_2_ concentration at room temperature is few seconds as reported by several authors [1,6,8,33]. Xie et al. achieved to obtain a response time of 0.7 s for 2.2% of H_2_ in N_2_ at room temperature with a sensor response of 600% [33].

For the studied devices, the mechanism of sensing in a neutral gas environment is closely dependent on the volume change property of Pd upon H_2_ exposure. Indeed, formation of palladium hydride (PdH_x_) occurring when H_2_ is adsorbed and diffused into Pd lattice leads to a volume expansion of Pd NPs (Figure 7d, [6]). The related structural phenomenon is described earlier in the present document. This structural change induces a close of gaps between NPs allowing an increase of the sensor conductance (decrease of resistance) [1,6,33]. Indeed, resulting from the increase of volume, the NPs touch each other reaching thus the percolation threshold. The changes in resistance or conductance are measured and gives rise to the sensor response. Figure 7 from [6] shows all the steps of this sensing mechanism. A slight increase in resistance can also be observed at an earlier stage of gas exposure due to the formation of few PdHx species that are known to have a higher resistivity than Pd. When bigger amounts of hydrogen are absorbed, the decrease in resistance due to volume expansion as described before takes precedence over the PdHx-induced resistance increase [6]. As observed by Gupta et al. [6], it should be noted that the PdH_x_-induced resistance increase is dominant when selecting air as carrier gas for hydrogen sensing experiment. Thus, only an increase in the resistance device is observed upon H_2_/air exposure. In fact, when working under air environment, the partial pressure of oxygen is higher than hydrogen partial pressure leading to more oxygen species, thus inducing competitivity between H_2_ and O_2_ adsorption. Therefore, only low H_2_ adsorption is observed leading to small volume expansion and therefore only increase in resistance due to formation of few PdH_x_ species takes place. Gupta’s team found out that for the studied concentrations, responses due to volume change are relatively faster than those only induced by resistivity change from PdH_x_ formation. Response time of sensor working under air was therefore higher than in N_2_ background experiment [6]. 

##### Pd NPs on Other Oxide Materials 

Several teams have also investigated hydrogen sensors with a configuration of Pd NPs on various metal oxide (MO_x_) structures [54,59,60,72]. Examples of devices working with Pd NPs-decorated ZnO [55,58,60], WO_3_ [51,52,53,61,62,64], TiO_2_ [56,57], NiO [54] and SnO_2_ [49,65], among other MOx structures, are provided in the literature. Typically, while Pd NPs act as a catalyst in adsorption and dissociation of gas molecules, metal oxide structures ensure the electron conductive path.

It should be emphasized that a quite common sensing mechanism is observed for all the sensors described in this section. Figure 8 [51] illustrates this working principle. 

This process might be associated to Pd-based MO_x_ sensors as reported in [53]. In an initial stage, before hydrogen exposure, ambient oxygen molecules are adsorbed and dissociated by Pd NPs. Due to a spillover effect, they diffuse onto the MO_x_ surface, from where they would capture free electrons resulting in ion species (Equations (4)–(6)) [63] and formation (or enlargement) of a depletion layer. In the presence of hydrogen, H_2_ gas molecules are similarly adsorbed and dissociated by the metal nanoparticles. Once again, resulting from a spillover effect, they diffuse onto the surface of the MO_x_ where they react with pre-adsorbed oxygen ions (Equations (7)–(9)) [63]. Resulting from this interaction, water vapor molecules are formed and electrons are released reducing the depletion layer and increasing the device conductance. [49,59,63]. In the case of p-type semiconductor, the released electrons recombine with holes at the MO_x_ surface therefore increasing the device resistance [54]. 

(4)$$O_{2\left(\mathrm{gas}\right)}\to 2O_{\left(\mathrm{adsorbed}\right)}$$

(5)$$O_{\left(\mathrm{adsorbed}\right)}+e_{\left({\mathrm{from}\;\mathrm{MO}}_{x}\right)}^{-}\to O^{-}$$

(6)$$O_{\left(\mathrm{adsorbed}\right)}+2e_{\left({\mathrm{from}\;\mathrm{MO}}_{x}\right)}^{-}\to O^{2-}$$

(7)$$H_{2}\to 2H_{\left(\mathrm{adsorbed}\right)}$$

(8)$$2H_{\left(\mathrm{adsorbed}\right)}+O^{-}\to H_{2}O+e^{-}$$

(9)$$2H_{\left(\mathrm{adsorbed}\right)}+O^{2-}\to H_{2}O+2e^{-}$$

A small variant of the mechanism suggests that upon hydrogen exposure, these gas molecules are absorbed by Pd NPs and formation of Pd hydride occurs. The hydride reacts with the adsorbed ionic oxygen following Equation (10) and free electrons are released thus increasing the conductance [56,57]. Nonetheless, this mechanism is also based on the interaction between H species and oxygen ions that releases electrons to vary the resistance properties of the system:(10)$${\mathrm{PdH}}_{x}+\frac{x}{4}O^{2-}\;\left(\mathrm{ads}\right)\to \mathrm{Pd}+\frac{x}{2}H_{2}O+\frac{x}{4}e^{-}$$

Examples of works implementing Pd NPs on MO_x_ materials for H_2_ detection applications are given in this section with a focus on the best sensing performances achieved in term of limit of detection and time response:

##### Pd NPs on ZnO structures

T-Roksana Rashid et al. achieved to obtain a good sensor response with H_2_ concentrations down to less than 0.5 ppm at room temperature. A response time of 18.8 s was found when working under a gas concentration of 1000 ppm with a sensor response of 91%. These performances were achieved with an experimental detection system working with Pd NPs sputtered onto Ga-doped ZnO nanorods grown on polyimide tape to ensure flexibility of the device. Experiments were performed using N_2_ as carrier gas [47].

##### Pd NPs on NiO structures 

Sta et al. implemented a sensor for H_2_ detection working with a NiO film partially covered by Pd NPs [54]. Sensing experiments were carried out selecting air as carrier gas and investigating the sensor response in a temperature range from 53 °C to 180 °C. The sensor detection range was from 1000 (0.1%) to 15000 (1.5%) ppm. The optimum response of the device was found at 140 °C where the sensor exhibited a signal of 14% for a hydrogen concentration of 10000 ppm (1%) and a response time of 3 min.

##### Pd NPs on TiO_2_ structures

The team of Xiang reported performances of a room-temperature hydrogen sensor working thanks to Pd NPs-doped TiO_2_ nanotubes. The sensor was fabricated by electrochemical anodization of a titanium foil resulting in the formation of TiO_2_ nanotubes and followed by reduction of Pd precursor on the surface of nanotubes through wet chemical methods. For sensing experiments, air was selected as carrier gas and the device showed detectable response in a range of hydrogen concentration from 1% to 5%. An average response time of 2 min was found [56].

##### Pd NPs on In_2_O_3_ structures 

The hydrogen sensing properties of In_2_O_3_ functionalized with Pd NPs have been investigated by Liu et al. [57]. They developed a device working with a Pd NPs-decorated In_2_O_3_ film and deposited onto an alumina substrate equipped with interdigitated Au electrodes. The sensing experiments were performed at room temperature within a detection range of H_2_ from 0.05% to 3% in air. At 1% H_2_, they achieved to obtain a response time of 28 s.

##### Pd NPs on SnO_2_ structures

Li et al. [49] developed a H_2_ detection device working with Pd NPs-doped SnO_2_ microspheres at 200 °C in air. They achieved to detect H_2_ concentration down to 10 ppm. They performed experiments up to 3000 ppm and obtained good sensor answer in the whole concentration range. A response time of 4 s at 3000 ppm (0.3%) of H_2_ was obtained [49]. Another sensor using Pd NPs-doped SnO_2_ materials is described in the work of Zhang et al. [50]. They fabricated a device based on Pd NPs-SnO_2_ nanofibers by electrospinning method. It presented a limit of detection around 4.5 ppm and the response and recovery times were about 9 s at 100 ppm of H_2_ in air when the temperature was set at 280 °C.

##### Pd NPs on Mn_2_O structures

Sanger et al. [63] implemented a H_2_ detection device based on Mn_2_O nanowalls decorated by Pd NPs and responding to the as-described sensing mechanism. The sensor could detect hydrogen concentration as low as 10 ppm and the sensing experiment were performed from 10 to 1000 ppm H_2_ at 100 °C. A response time as low as 4 s was found at a concentration of 100 ppm H_2_ [63].

##### Pd NPs on WO_3_ structures 

As mentioned by Wang et al. [64], it is noteworthy that an alternative sensing mechanism might occur during H_2_ detection experiments with these sensing materials. They suggested that the adsorbed hydrogen reacts with oxygen from the lattice of tungsten oxide resulting in formation of water and free electrons in one hand. In another hand, this reaction might also lead to partial reduction of WO_3_ and forms hydrogen tungsten bronzes H_x_WO_3_ according to Equation (5) [53,64]. The formation of hydrogen tungsten bronzes revealed by a coloration phenomenon was also observed elsewhere [53].

(11)$${x\;H}_{\mathrm{ads}}+{\mathrm{WO}}_{3}\to H_{x}{\mathrm{WO}}_{3}$$

Working with this kind of sensors, Kabcum’s team obtained a ultra-high response of 3.14 × 10^6^ at 3% of H_2_ with a response time as low as 1.8 s at 150 °C [52]. Pd NPs were impregnated onto WO_3_ powder. A paste was made from the nanopowder and was spin coated onto alumina substrates equipped with interdigitated Au electrodes. Air was used as carrier gas for sensing experiments. Moreover, Annanouch et al. [51], working with p-type PdO nanoparticles supported on n-type WO_3_ nano-needles gathered good sensor response at a hydrogen concentration as low as 40 ppm in H_2_/air mixture. Other interesting performances are obtained by Zhao et al. [73] who sputtered a dense Pd NPs layer on a highly porous WO_3_ film and investigated the H_2_ sensing properties of these materials. They achieved to obtain a very low response time of less than 1 s at 2% of H_2_ when working at 80 °C.

In view of the above examples, it is noticeable that for Pd NPs on MO_x_ systems, very different sensor performances could be obtained depending on the selected MO_x_, the fabrication process, the sensing parameters, etc.

##### Pd NPs on Carbon Material

Electrical Pd-based sensor for hydrogen detection also took advantage of the development of carbon materials (graphene, carbon nanotubes, etc.) these recent years. Several teams have worked on the implementation of such devices [9,39,46,66,67,68,69,70,71]. Typically, in this configuration, Pd NPs decorate graphene sheets or carbon nano-tubes or even nanowires and the resulting structure is implemented on top of SiO_2_/Si or glass substrates as shown in Figure 9a [39] and Figure 9b [39,46,67,70,71]. Nonetheless, other substrates [46,66] or specific configurations [68,69] have also been reported by a few teams. For example, Chung et al. [46] deposited Pd NPs-decorated graphene on polyethylene terephthalate to ensure flexibility of the fabricated device and Seo et al. [69] developed sensors composed of Pd NPs deposited on single suspended carbon nanowires (Figure 9c [69]). In the systems for hydrogen detection depicted in this section, metallic electrodes are used to connect the sensing sample to a multimeter in order to measure the electrical change upon H_2_ exposure. This type of devices has attracted a lot of interest, with researchers investigating different carbon materials and Pd NPs synthesis and deposition technique (thermal evaporation technique, electrochemical deposition, microwave irradiation technique, etc.), different particle sizes, density and morphology, different positioning of electrodes as well as experimental parameters [9,39,46,66,67,68,69,70,71].

When implementing this type of sensors, detection of hydrogen is generally studied at room temperature [9,39,46,66,67,69,70,71]. Phan’s team working with cubic Pd NPS deposited on graphene sheets also investigated the response of their device at 50 °C and 100 °C. The sensor response decreased with the increase of temperature. At 1000 ppm of H_2_, the response values were 13%, 10.4%, and 9.2% at room temperature, 50 °C, and 100 °C, respectively. However, as the temperature increased, the sensor demonstrated better (lower) response and recovery times because hydrogen molecules are easily absorbed or desorbed inducing faster response and recovery times. At 50 °C, optimal linearity and repeatability in the sensor response were achieved [67].

The sensors investigated in this section usually work using air as carrier gas [9,66,67,68,69,70,71] likely for more simplicity of the setup to be implemented. Nonetheless, the use of neutral carrier gas is also reported. Kumar et al. selected argon (Ar) as carrier gas when sensing H_2_ with Pd NPs thermally evaporated on a graphene sheet [39]. Chung’s team implemented a setup involving Pd NPs coating a graphene layer and the hydrogen detection tests were performed using N_2_ as carrier gas [46]. The use of neutral gas reinforces the exclusion of the other reactive gas molecules effect, especially oxygen [46]. The detection range of the present systems commonly lies in the low H_2_ concentration range, typically from less than 10 ppm up to concentrations around 5% [39,69,70] making those devices good candidates for leak detection. Phan et al. working with Pd NPs on graphene sheets as sensing material reported a limit of detection as low as 0.2 ppm H_2_ in N_2_ at room temperature [74].

By working with the present devices in low range H_2_ concentrations, it is generally observed that when the gas concentration increases, the sensor response increases, and response times decrease, as seen in Figure 10. Typical response times at the studied gas concentration range (less than 5%) at room temperature are about few tens of seconds [9,46,66,69,70,71]. Nonetheless, the team of Kumar worked with Pd NP-decorated graphene layers and achieved a response time of 6 seconds with a sensor response of 51.4% at a H_2_ concentration of 2% [39]. Sun et al. [75] even reported a response time of 3 s at 1% H_2_ concentration with a response of 130% when using Pd NP-decorated single-walled carbon nanotubes as sensing material.

The sensing mechanism of the devices investigated in this section results from the influence of hydrogen on interaction between Pd and carbon material. Graphene as carbon nanotubes are well-known to contain p-type carriers (holes) in their structures. When Pd NPs-graphene/CNT structures are exposed to hydrogen gas, these last molecules dissociate at the Pd surface, then are adsorbed and they diffuse into the Pd lattice leading to the formation of PdH_x_. An electron transfer from PdH_x_ to graphene (or CNT) occurs inducing recombination with holes in the carbon material thus increasing the resistance of the latter [9,39,46,66,67,68,70].

PdH_x_ has a lower work function compared to both Pd and graphene (or CNT) and the electron transfer from PdH_x_ to graphene (or CNT) is then favored [39]. The variation of the system resistance is measured and gives the sensor response. Moreover, Alfano et al. reported that a decrease in resistance might occur when the hydrogen concentration increases due to the transition from α-PdH_x_ to β-PdH_x_ at higher partial pressure of H_2_ and knowing that those two species possess distinct chemical and physical properties. It is worth noting that this phenomenon is dependent on the Pd amount involved in the sensing process [66].

##### Pd NPs on specific substrates

In addition to all the Pd NPs-based H_2_ electrical sensor families investigated in the previous sections, the literature also describes the use of some specific substrates developed by very few teams [38,72,76] that display good performances regarding the device response time and limit of detection. For instance, in the work of Raghu et al., Pd NPs are dispersed on a graphitic carbon-nitride (gC_3_N_4_) layer supported by an alumina substrate to ensure H_2_ detection in a concentration range from 1% to 4% in air at room temperature. A sensor response of 99.8% for 4% H_2_ was achieved within a response time of 88 s. The sensor was able to operate up to 80 °C without significant changes in its performance [38]. The sensor response arises from the decrease in device resistance in the presence of hydrogen gas molecules. Upon H_2_ exposure, hydrogen molecules are adsorbed and dissociated on Pd surface where this interaction induces formation of palladium hydride. PdH_x_, as mentioned previously, has a lower work function compared to Pd. As a consequence, transfer of electrons takes place from PdH_x_ to gC_3_N_4_ increasing the charge carrier concentration in gC_3_N_4_ therefore leading to a fall of resistance [38]. Another example is the hydrogen sensor developed by Baek’s team where Pd NPs functionalize layers of MoS_2_ [72]. With their device, they were able to to detect H_2_ concentrations down to 50 ppm in air at room temperature. The sensor working principle is similar to the one described for Raghu et al.’s device. This sensing mechanism is illustrated in Figure 11 [72].

Working with Pd NPs-decorated nanoporous poly(aniline-co-aniline-2-sulfonic acid):poly(4-styrenesulfonic acid) (P(ANI-co-ASA):PSS), Cho et al. [77] reported the fabrication of a hydrogen sensor with a limit of detection of 5 ppm H_2_ in air at room temperature and presenting 90 s and 40 s for respectively response and recovery times. While the characteristic detection times still must be improved, the limit of detection obtained in this work is quite low and therefore very promising for leak detection applications.

##### Conclusion on Electrical Sensors

Electrical hydrogen detection systems working thanks to Pd NPs have been and are still widely investigated by many research teams all over the world. Table 3 summarizes the performance of the most notable of these experimental devices. Only devices with either a response time of ≤10 s or a limit of detection ≤ 100 ppm have been selected in this table. Xie et al. achieved to obtain a response time of 0.7 sec in N_2_ at room temperature with a sensor answer of 600% for 2.2% of H_2_ [33]. Their device was composed of closely spaced Pd NPs deposited onto a SiO_2_ substrate equipped with interdigitated electrodes. In terms of response time, the performance of their sensor is remarkable. However, even if the device could give good response at H_2_ concentrations as low as a few tens of ppm (which is much lower than the explosion limit for H_2_ in air (4%)), better limits of detection have been reported elsewhere. Phan et al. succeeded in obtaining a good sensor response at a hydrogen concentration as low as 0.2 ppm in N_2_ at room temperature [74]. To the best of our knowledge, it is the lowest limit of detection reported so far for Pd NPs-based hydrogen electrical sensors. They fabricated a H_2_ sensor with Pd NPs on graphene sheets as sensing material. The best response time they obtain was few minutes at 1000 ppm H_2_ which is more than the sub-second response time in Xie’s work [33]. The best device performances reported so far, in terms of response time and hydrogen limit of detection respectively, are ascribed to two distinct works. Nonetheless, the device proposed by Xie et al. [33] seems to present a better compromise in term of response time and limit of detection performances.

It can be observed that globally, sensors fabricated by direct deposition of Pd NPs on Si or SiO_2_ substrate give very low response time (typically few seconds) and work in an acceptably low range of H_2_ concentrations (0.1% to 5%), while the H_2_ sensor configuration of Pd NPs on carbon material is able to detect, with good sensor response, very low hydrogen concentrations (less than 10 ppm) and shows typical response time of a few tens of seconds. For Pd NPs on MO_x_ systems, sensor performances are really different from one another due to the selected MO_x_, the fabrication process, the sensing parameters, etc. Figure 12 present a schematic of the proportions of high performances in limit of detection reported in the literature for electrical Pd NPs-based hydrogen sensors. In most cases, extremely low LODs (less than 10 ppm) are achieved when selecting the configuration of Pd NPs on carbon material as sensing sample. As will be discussed later, it is also seen that more and more Pd-based bimetallic structures are selected to achieve those high performances.

It should be pointed out that two different behaviors of Pd NPs can be distinguished in the sensing mechanisms of the devices investigated here. First, when Pd NPs are directly deposited on Si or SiO_2_ substrate, the metallic particles act as the sensing materials ensuring a conductive electron path by closing the gaps between particles due to volume expansion upon H_2_ exposure [1,6,33]. Second, when Pd NPs are combined with carbon materials or metal oxide structures, the last mentioned ensure the electron conductive path and the role of the metallic particles is to act as a catalyst in adsorption and dissociation of gas molecules [9,39,46,49,54,56,58,59,60,65,66,67,68,70,72].

One way of improving Pd NPs-helped electrical detection of hydrogen is to use bimetallic nanoparticles. They can be palladium-based alloy or core@shell nanoparticles. As seen in Table 2, the literature provides several examples of such structures. Pd-Ni nanoparticles have been widely investigated [78,79,80,81] because the addition of Ni was found to induce a large change on the α- β phase transition behavior [79,81], reduce the hysteresis behavior of the Pd NPs [78] and hence enhance the performances of nanoparticle-based sensors. Phan et al. [78] reported a limit of detection lower than 10 ppm for Pd-Ni alloy NPs supported by graphene with a better sensor response at this H_2_ concentration range when compared to the use of pure Pd NPs at room temperature. Sun et al. [79] found lower sensor response times for Pd-Ni NPs when comparing the use of pure Pd NPs and Pd-Ni NPs on a silicon substrate at room temperature. The use of Pd-Pt NPs has also been reported many times [55,82,83,84]. Indeed, platinum is also known to have a good affinity with hydrogen. Kumar et al. [83] fabricated a device with Pd-Pt alloy NPs deposited on graphene layers and achieved a response time as low as less than 2 s at 2% H_2_ concentration. Moreover, Hassan et al. [55] mentioned that especially the use of Pd-Pt core@shell NPs would preferably induce individual sensing properties of the two elements rather than alloying phenomenon, thus improving the device performances. They reported the use of Pd@Pt core@shell NPs on ZnO nanorods which allowed a detection of H_2_ concentration of only 0.2 ppm and the device showed a response time of 5 s at 1% of H_2_. There are many other Pd-based bimetallic NPs investigated for electrical detection of hydrogen. For example, Au@Pd core@shell NPs have been used in [85] because of the easiness of the sample preparation and the team of Sharma [86] reported investigation on Pd-Ag alloy NPs that might induce less H_2_-generated embrittlement problems compared to pure palladium. Furthermore, the use of similar hybrid structures such as Pd@C core shell NPs [87] and Pd-polyelectrolyte hybrid NPs [88] is also investigated in the literature for electrical detection of H_2_ in order to obtain specific properties. To the best of our knowledge, the performances of those later hybrid nanostructures are not yet comparable to those described in this section in term of response time and limit of detection, however further improvement still have to be done.

Despite the good performances recorded for electrical hydrogen detection systems working by use of Pd NPs, some issues still need to be faced with this type of sensor [15,89,90]. Safety and sensor longevity issues due to the use of electrical contact in harsh environment are major challenges that these electrical sensors have to overcome [89]. One way to achieve this goal is the use of optical sensors. In the next section, the use of Pd NPs-based optical hydrogen sensors will be discussed.

**Table 3 sensors-19-04478-t003:** Resume of best performances Pd NPs-based electrical hydrogen gas sensors found in the literature. Unless specified, the sensor response is obtained by the relation given in Equation (3). (*) Sensor response is given by the ratio of the resistance in air and in the hydrogenated environment. (**) Sensor response is given by the relative variation of conductance. (***) Sensor response is given by the relative variation of current intensity. (****) Sensor response is given by the relative variation of resistance defined as (R_O_ − R_H2_)/R_H2_, where Ro is the base resistance of the sensor in air, and R_H2_ is the resistance measured in a H_2_-containing environment.

Sensing Material	Limit of Detection	Response Time	Recovery Time	T°	Sensor Response	Ref.
Pd NPs	1%	1.2 s/1%	10 s	RT	96% ^(^**^)^	[8]
Pd NPs	0.1–1%	3 s/0.1%	31 s	323 K	15.40%	[6]
Pd NPs	few ppm	0.7 s/2.2%	10.2 s	RT	600% ^(^**^)^	[33]
Pd NPs/Si nanowires	nr	5 s/1%	nr	RT	3400% ^(^***^)^	[91]
Pd NPs/Si nanowires	nr	3 s/5%	nr	nr	nr	[92]
Pd NPs/SnO_2_	10 ppm	4 s/0.3%	10 s	473 K	315.34 ^(^*^)^	[49]
Pd NPs/SnO_2_	4.5 ppm	~9 s/0.01%	~9 s	553 K	8.2 ^(^*^)^	[50]
Pd NPs/WO_3_	~1 ppm	1.8 s/3%	nr	423 K	3.14 × 10^6 (^*^)^	[52]
Pd NPs/WO_3_	40 ppm	120 s/0.05%	12 min	473 K	10^3 (^*^)^	[51]
Pd NPs/WO_3_	10 ppm	10 s/0.005%	50 s	403 K	382 ^(^*^)^	[93]
Pd NPs/WO_3_	<500 ppm	7 s/2%	nr	RT	230 ^(^*^)^	[53]
Pd NPs/WO_3_	nr	<1 s/2%	50	353 K	>10^4(^****^)^	[73]
Pd NPs/ZnO	<0.5 ppm	18.8 s/0.1%	nr	RT	91.2%	[47]
Pd NPs/MnO_2_	10 ppm	4 s/0.01%	~20 min	373 K	11.5 ^(^*^)^	[63]
Pd NPs/MoS_2_	50 ppm	780 s/1%	~15 min	RT	35.3%	[72]
Pd NPs/SWCNTs	10 ppm	7 s/1%	nr	RT	25%	[48]
Pd NPs/SWCNTs	250 ppm	5 s/2.5%	nr	RT	nr	[94]
Pd NPs/SWCNTs	100 ppm	3 s/1%	5 min	RT	130%	[75]
Pd NPs/Graphene	0.2 ppm	~240 s/0.1%	~600 s	RT	7%	[74]
Pd NPs/Graphene	200 ppm	6 s/2%	45 s	RT	51%	[39]
Pd Ncubes/Graphene	6 ppm	nr	nr	323 K	10.4%	[67]
Pd NPs/CNTs	<5 ppm	~30 s/0.1%	~3 min	RT	~7.3%	[68]
PdNi NPs/Graphene	<10 ppm	180 s/0.1%	720 s	RT	11%	[78]
PdPt NPs/Graphene	nr	< 2 s/2%	18 s	313 K	4%	[83]
Pd@Pt NPs/ZnO nanorods	0.2 ppm	5 s/1%	76 s	373 K	58%	[55]
Pd@Pt NPs/Si nanowires	<10 ppm	7.7 s/1%	7.7 s	348 K	5.02%	[82]
Pd NPs/C nanowires	10 ppm	< 70 s/10 ppm- 1%	5 s	RT	175%	[69]
Pd NPs in Nafion	~0.2%	10 s/nr	nr	RT	nr	[76]

#### 2.2.2. Optical Sensors

In this section, optical sensors for detection of hydrogen gas molecules involving the use of Pd NPs will be investigated. It must be noted that literature mostly provides results from Pd films-based compared to Pd NPs-based optical H_2_ sensors. Due to the increased surface to volume ratio, the use of Pd NPs may provide improved performances resulting from higher effective surface available for interaction of Pd with the hydrogen gas molecules [15]. The use of nanoparticles has several advantages, such as the possibility to engineer the sensor response time owing to materials design and also by tailoring the particles dimensions to reduce gas diffusion times [11].

A simple and typical experiment set up for this kind of sensors, as used by Corso et al. [15], is illustrated in Figure 13. Typically, an experiment is performed by measuring the change in optical properties (transmittance, absorbance, reflectance, etc.) of the sensing material when switching from the carrier or recovery gas to H_2_ enriched environment in a cyclic manner inside a gas chamber [10,11,12,15,90,95]. A light source is used to illuminate the sensing sample and the optical response of this latter within a set environment in the gas chamber is recorded by a photodetector.

In several cases, α to β phase transition during Pd-hydride formation is an important parameter in the detection mechanism [10]. This arises from the fact that the optical constants of the palladium hydride evolve from α to β phase and therefore its optical properties [12]. However, the optical response of the α-phase usually does not differ strongly from the pure metal-associated response [12,35]. Therefore, the sensitivity (seen as contrast in response between hydrogenated and non-hydrogenated state) of the sensing sample to the α-phase of PdH_x_ is often challenging [12,96]. Sensing samples sensitive to the α-phase PdH_x_ formation are able to detect low H_2_ concentrations and the sample sensitivity to β-phase PdH_x_ allows detecting higher concentrations of hydrogen (typically higher than ~1%) [12,15].

As reported by many authors [10,12,15,90,95], another key parameter is the synthesis process of the Pd NPs because it influences the morphology, structure and organization of the Pd NPs which result in specific optical properties upon sample exposure to hydrogen. In this context, for example, Corso et al. [15], Kracker et al. [90], and Isaac et al. [95] reported an initial hydrogenation-dehydrogenation cycle in order to stabilize the sensing Pd NPs samples. Ohodnicki et al. [89] also mentioned this pretreatment technique to palliate the influence of systematic partial oxidation of Pd and AuPd alloy NPs in optical sensing of hydrogen.

In the same way as electrical Pd NPs-based H_2_ sensors have been sorted in the previous section, optical Pd NPs-based H_2_ sensors will also be distinguished here regarding the configuration of the sensing sample. Hence, sub-sections investigating optical Pd NPs-based H_2_ sensors taking advantage of properties of Pd NPs on Si-based substrates (fused silica, glass substrate, quartz foil…) as well as on optical fiber or decorating specific waveguides will be developed in the following sections.

##### Pd NPs on Si-based substrate

Here, the sensing sample is composed of a Si-based substrate coated by Pd NPs. With this configuration, several authors also used bimetallic Au/Pd NPs to take advantage of the well-defined plasmonic resonance of Au [10,11,95]. Therefore, this section will first focus on the use of Pd NPs for this kind of sensor, and then the use of bimetallic Au/Pd NPs will also be addressed.

##### Pd NPs

Interesting performances have been reported for this type of sensors [12,15,90]. Recently, Corso et al. [15] developed an optical device for H_2_ detection which gave a response time as low as 2 s and a recovery time of around 5 s at a 5% hydrogen concentration in nitrogen. The experiments were carried out at room temperature. On fused silica substrates, they deposited thin Pd layers thanks to e-beam evaporation. The Pd layer was oxidized and then underwent a reducing irreversible treatment under a first prolonged exposure to hydrogen that resulted in the formation of Pd nano-grains. The sensor response, also identified as the optical absorbance change (OAC), was evaluated by the difference between the absorbance of the sample in the hydrogenated environment and that in the dry air atmosphere. The optical absorbance in each environment was in fact calculated as the integrated absorbance values over the whole spectral range from 400 nm to 800 nm. At the studied H_2_ concentration of 5%, the change in Pd optical response was due to the formation of the palladium hydride in β-phase resulting from the splitting of the H_2_ molecules at the metal surface and diffusion of the hydrogen atoms into the Pd lattice. Although Corso’s work [15] allowed them to obtain quite low response times (2 s) for the sensing of H_2_, the mainly investigated concentration of 5% H_2_ still remains above the lower explosive limit of H_2_ in air (4%).

Watkins et al. [12] recently reported the implementation of a nanoparticulate Pd film on a glass substrate as a sensing material for H_2_ detection that showed a limit of detection as low as less than 10 ppm of H_2_ in Ar at room temperature. This sensor sensitivity is comparable to the best result obtained with electrical devices. In the work of Watkins’s team, agglomerated Pd NPs that form the sensing film are obtained by oblique angle vacuum deposition and it appears that they are mostly oriented close to the direction normal to the evaporation orientation. Therefore, to both light polarizations either parallel or perpendicular to the direction of evaporation, the sample showed a dichroic response due to localized surface plasmon resonance (LSPR) within the nanoparticles forming the film. The device response was monitored by measuring the transmission anisotropy (TA) and was evaluated by the difference between the TA of the sample in the hydrogenated environment and that in Ar at the given wavelength of 885 nm. The change in TA upon H_2_ exposure resulted from the formation of palladium hydride and thus a change in the dielectric function of Pd. A wide range of H_2_ concentration from pure hydrogen to 2 ppm was investigated as shown in Figure 14 [12]. Three response behaviors corresponding to the formation of the β-phase PdH_x_ at high H_2_ concentration (4–100%), the α-phase at very low H_2_ concentration (0.2% and below) and both α-phase and β-phase around 1% were found. The measurement of optical anisotropy was even sensitive to the formation of the α-phase PdH_x_ that occurs at very low level of hydrogen and therefore allowed detecting very small H_2_ concentration down to 10 ppm. However, the smallest reported response times at low H_2_ concentrations were about a few tens of seconds (more than the 2 s reported in [15]).

To improve and better control the performances of this type of sensors, some researchers have reported the use of bimetallic Au/Pd NPs.

##### AuPd NPs

Bimetallic Au/Pd NPs on Si-based substrate for optical detection of H_2_ have been investigated either on the configuration of core@shell nanoparticles [10] or metallic alloys with different Au content [11,95]. Song et al. [10] investigated core@shell nanoparticles deposited on quartz substrates with an Au core and a Pd shell for optical sensing of hydrogen. The use of an inert Au core could decrease the diffusion length of hydrogen in the Pd shell while controlling the thickness of the Pd shell allows adjusting the device optical response properties. Measuring the reflective response of the particles illuminated by a LED centered at 625 nm, they achieved to detect H_2_ concentration as low as 0.1% with Au cores of 40 nm and 22 nm thick Pd shells at room temperature. Decreasing the Pd shell from 22 nm to 4 nm allowed reducing the response time from 62 s to a value as low as 4 s at 4% H_2_ concentration. They found a complete positive correlation between the decrease in thickness of Pd shell and the reduction in response/recovery time of the sensor. The measured change in reflectance depended on hydrogenation and dehydrogenation of the Pd shell as the hydrogen exposure resulted in a reduction of the volume density of free electrons, thus inducing a decrease in the permittivity of Pd.

Apart from the use of core@shell Au@Pd NPs, bimetallic alloy nanoparticles of Au-Pd have also been investigated for optical detection of hydrogen gas molecules [11,95]. Alloying Pd with Au may give more accuracy in the readout signal of the sensor [11]. In fact, as reported by Wadell et al. [11], the presence of Au content in the sensing metallic NPs induces a shrinkage of the hysteresis effect during hydrogenation and dehydrogenation cycles as a result of a decrease in the hydrogen binding energy and/or the attractive hydrogen−hydrogen interaction in the system when the palladium hydride formation is thermodynamically favorable. Figure 15a [11] illustrates this phenomenon. This hysteresis, as also mentioned by Isaac [95] is undesirable. It is responsible of unwanted uncertainty in the quantitative readout of a hydrogen sensor signal because, due to it, the signal might not only depend on the current hydrogen pressure but also on the hydrogenation/dehydrogenation treatment already undergone by the material. The hysteresis is material dependent and not sensor readout dependent; therefore tailoring the sensing material (for example by alloying Pd to Au) allows reducing it [11].

Another beneficial effect of the use of Au-Pd alloy NPs could be the possibility to engineer the response time of the sensor as shown in Figure 15b [11]. Wadell et al. [11], for a varying Au content in the sensing metallic NPs from 0 to 25%, found a decrease in the response time as the Au content in the alloy was increased. They reported high sensor performances with a response time as low as less than 1 sec in a H_2_ concentration range from less than 0.1% to 4% at 30 °C. However, the experiments were carried out in vacuum and therefore might not represent what may happen in a real ambient environment. An unpolarized white light was sent to the sample in a vacuum cell and the optical transmittance of this latter upon H_2_ exposure was recorded. The sensor response was given by the change in the LSPR peak intensity in the presence of hydrogen.

##### Pd NPs on optical fiber 

The use of optical fiber for H_2_ detection based on Pd have been widely investigated since 1984 when Butler developed the first optical fiber sensor for hydrogen detection [96,97,98]. The device response arose from a phase modulation of light source upon hydrogenation of the sensing Pd sample. As reported by Javahiraly [96], optical fiber hydrogen sensors based on Pd could be sorted as a function of the sensor response that can be phase-modulated, polarization-modulated, intensity-modulated or wavelength modulated. Detection and measurements of hydrogen concentration result from either the sensitivity to palladium hydride expansion (phase-modulated, wavelength modulated), the change in Pd refractive index (intensity-modulated) or the heat release from the chemical reaction between Pd and hydrogen (polarization-modulated). A lot of works has been reported for H_2_ detection by means of optical fiber functionalized by Pd film layers as sensing material. Nevertheless, the use of Pd NPs instead of Pd layers as one promising direction for the discussed application has also been reported in [96]. Advances in this direction will be addressed in this section.

It is noteworthy that, in order to meet the most important requirements (high sensor response, short response and recovery times, low limit of detection, etc.) for high performance sensors, an interesting and often used proposal is the use of tapered optical fiber (TOF) [17,89,99,100,101]. This results from the intense evanescent field generated in the tapered fiber region [99,102,103] and which is highly sensitive to its surroundings. Therefore, a TOF-based sensor is dependent on the evanescent wave absorption mechanism [99]. Tapering the dimensions of a conventional optical fiber induces the enhancement of the evanescent field since a lot of transmitted signal into the TOF spreads out the cladding as an evanescent field. As reported by Gonzalez-Sierra et al. [99], it is expected that the sensor response relies on the intensity of evanescent field present at the physical boundary of the TOF. As a consequence, the evanescent field is affected whenever any disturbance in the environment surrounding the TOF occurs and the output signal of the optical sensor is thus modified [99,104].

##### Pd NPs

Recently, Li et al. [17] reported the use of Pd NPs embedded in an amorphous PMMA thin film which coated a silica microfiber for optical detection of hydrogen. The composite sol-gel PMMA was used to disperse and elaborate metallic nanoparticles on the surface of the silica microfiber whose cladding layer had previously been removed. H_2_ concentrations in the range of 0.2 to 1% in nitrogen were investigated and the lowest response time was found to be around 5 s. A sketch of the experimental set up is presented in Figure 16a [15]. As a light source, an amplified spontaneous emission light with a wavelength spectrum from 1520 to 1560 nm was used and the transmission spectra of the microfiber were recorded.

As observed in Figure 16b, resulting from excitation by the evanescent wave, a red shift of the resonance peak occurs upon hydrogenation due to the formation of PdH_x_ species that induces a decrease in the refractive index of the composite film [17].

Although sensor performances reported in the work of Li et al. [17] are not always reached, several researchers have worked on this type of device and interesting results have been reported. As a matter of fact, Poole et al. [101] reported the implementation of a Pd-nanoparticles infused mesoporous-TiO_2_ film integrated optical D-fiber which makes possible the detection of gradients of hydrogen concentration at high temperatures. This result is very promising for application such as the measurement of H_2_ concentration in fuel cells in which concentration gradients may exist due to variations in the fuel consumption all along the cell length. The response times were of the order of 2 to 3 min (far longer compared to the 5 s obtained by Li et al. [17]).

##### AuPd NPs

In a work reported by Monzon-Hernandez et al. [100], Au-Pd alloy NPs with about 35% of Au content were used to coat conventional telecommunication microfiber for optical detection of hydrogen. They were able to obtain, at room temperature, a response time of 2 s at 4% H_2_ concentration in N_2_ and a recovery time of 20 s. The single mode optical microfibers were tapered down to dimensions of the order of the wavelength of the light source and tests with non-tapered fibers did not give any signal. The gas concentration range studied was from less than 1% to about 8% where the sensor gave a saturated response. As a light source, a highly stable laser with wavelength of 1550 µm was used and the loss introduced by the light scattering on NPs was recorded through a single photodetector. When the wave-guided light is sent through the microfiber, the evanescent field interacts with the metallic nanoparticles resulting in a scattering out of the waveguide thus producing an increment of the transmission loss. The authors reported a pulsed-like response upon hydrogenation-dehydrogenation cycles that they attributed to a common result of two effects respectively optical and geometrical. Concerning the optical point of view, the exposure to H_2_ induces the formation of palladium hydride that results in a change of refractive index which in turn modifies the amount of scattered light and therefore the transmittance of the device. In parallel, hydrogenation also induces swelling of the AuPd NPs which leads to the increase of their size and probably the alteration of their shapes. The combination of both optical and geometric effects may give rise to the pulsed-like response.

It should be recalled that, as reported in [89], alloying Pd to Au might decrease the sensor response as compared to the use of pure Pd NPs. This is illustrated in Figure 17 [89]. The responses of these devices mainly result from the interactions of hydrogen with metallic palladium that induce the change of the metal properties. The addition of a significant Au amount would lower these interactions, for example through the decrease of the gas diffusion length in the palladium structure [10]. Therefore, the sensor answer is lowered. Ohodnicki et al. [89] investigated the response of an optical fiber sensor for hydrogen detection at different temperatures and using nitrogen as carrier gas. Pd and AuPd NPs embedded in SiO_2_ thin films were used to coat an unclad multimode silica-based optical fiber and interaction of the evanescent wave with the sensing elements upon hydrogen exposure when a light source (450–850 nm) was sent through the fiber induced the response of the sensor. H_2_ concentrations ranging from 2% to 100% were investigated. Lower responses were obtained in the case of AuPd NPs/SiO_2_ samples compare to Pd NPs/SiO_2_ samples and the authors attributed this observation to the reduced solubility of H_2_ in bimetallic Au-Pd materials.

##### Pd NPs on specific waveguides

Despite the fact that literature mostly mentions works presenting a configuration where Pd-based NPs coat either an optical fiber or a Si-based substrate for optical detection of hydrogen, few teams have also worked on alternative Pd NPs-based optical H_2_ sensor configurations.

Sirbuly et al. [105] reported the implementation of an optical H_2_ sensor using a suspended single crystalline SnO_2_ waveguide decorated with Pd NPs stabilized by octa(propylammonium)- polyhedral oligomeric silsesquioxanes (POSS) working at room temperature. Owing to their wide band gap (Eg = 3.6 eV) that allows transmission of photon frequencies out into the UV regime [106] and their refractive index (n~2.1) that allows their use in embedded applications due to their ability to confine and guide light in liquids, polymers, or glasses [107]. 

POSS-stabilized Pd NPs have a high affinity for attachment to SnO_2_ surface, their UV-vis absorbance spectrum presents excellent spectral overlap with the fluorescence from SnO_2_ waveguide and their optical response when exposed to H_2_ is very fast. For these reasons, they were selected in this work. The device showed high performances with a response and recovery times of the order of 1 and 2 s respectively and an experimental limit of detection of 0.5% of H_2_ in argon. The authors claimed that this sensitivity could be improved by further tuning sensing NPs sizes, waveguide dimensions, method of coupling light into the optical cavities, etc. The experimental set up is presented in Figure 18 [105]. White light was sent through the waveguide and after it travelled the sensing channel, it intensively scattered out of the cavity and was captured by a microscope objective and imaged with a quantitative camera. In the presence of hydrogen, the transmission of the waveguide is attenuated due to the change in refractive index of the sensing elements and interaction with the evanescent field as light travels through the waveguide. The response of the sensor was evaluated by the difference in optical transmission between signals obtained upon hydrogen exposure and without it over the initial transmittance level.

##### Conclusion on optical sensors

Interesting performances have been reported in the study of Pd NPs- based optical hydrogen sensors. Table 4 summarizes the performances of several of these experimental devices. 

Only devices with either a response time of ≤10 s or a limit of detection ≤100 ppm have been selected in this table. As discussed earlier in this section, in term of response time, the highest performances have been achieved in the work of Wadell et al. [11] with a reported response time of less than 1 s in a H_2_ concentration range from less than 0.1% to 4% at 30 °C. However, these results might not represent what may happen in a real ambient environment since the experiments were performed in vacuum. Apart from that, several authors have reported good results in H_2_/carrier gas mixtures with response times as low as less than 5 s [10,15,17,100,105]. In terms of sensitivity to the lowest hydrogen amount, remarkable results were obtained by Watkins et al. [12] who achieved the detection of very small H_2_ concentrations down to 10 ppm while for this type of devices most of the authors often do not report the detection of hydrogen concentrations lower than 0.1% (1000 ppm) [15,17,100,105].

Moreover, another interesting approach in the design of this type of sensor could be considered taking as reference the structures of Pd-based film materials already developed for H_2_ optical sensing and mimic the films configuration as nanoparticles. For example, the system developed by Perroton et al. [108] where a metal-insulator-metal (MIM) structure is deposited on the core of a multimode fiber and hydrogen is detected thanks to the spectral modulation of the light transmitted by the fiber could be a good candidate for this method (Figure 19 [108]). With their configuration, high performances were obtained: response times of 3 s at 4% hydrogen in argon. Furthermore, it is known that the use of Pd NPs may provide improved performances resulting from higher effective surface available for interaction of Pd with the hydrogen gas molecules [15] as well as the possibility to engineer the sensor response time by way of materials design and by tailoring the particles dimensions to reduce gas diffusion times [11]. Therefore, considering the use of core@shell NPs (Au@SiO_2_@Pd) taking as reference the MIM structure investigated by Perroton et al. [108] may allow achieving very high performances in hydrogen optical detection such as a response time below 1 s and improved sensitivity.

#### 2.2.3. Gasochromic Sensors

Pd NPs have also been used in the development of eye-readable sensors for detection of hydrogen. This type of sensors has been developed by only few teams as compared to the work reported for electrical and optical Pd NPs-based systems for hydrogen detection. Here, the sensing elements often consist on a structured metal (transition metal) oxide decorated by Pd NPs that act as catalyst for adsorption/desorption of hydrogen. The oxidation/reduction of the transition metal is responsible of the change in coloration [109,110,111,112].

At 4% H_2_, Kalanur et al. [110] obtained a response time of about 30 s with a gasochromic hydrogen sensor implemented by dispersing two dimensional (2D) WO_3_ nanostructures decorated with Pd NPs on a filter paper. Experiments were carried out at room temperature and the sensor showed a significant coloration change at hydrogen concentrations from 4% up to pure H_2_ where it only took 4 s to achieve a complete and reversible coloration change. In hydrogen enriched environment, the sensor coloration was modified from light-green-gray to dark-blue as it is observed in Figure 20 [110]. The proposed sensing mechanism is as follows: upon H_2_ exposure, the molecules of hydrogen are adsorbed and dissociated on the Pd NPs surface. Due to a spillover effect, the dissociated H atoms transfer into WO_3_ and react with oxygen species of the metal oxide structure which results in formation of water molecules and free electrons that are transferred to WO_3_. As a consequence, W^+6^ are reduced into W^+5^ which are responsible of the blue color [110]. Experiments were also performed at 1 and 0.1% H_2_ but only partial changes of color were observed, probably due to a small amount of tungsten ions involved during interaction. However, the authors reported that, at these hydrogen concentrations, their sensing materials could be used as efficient sensing elements for electrical detection of H_2_. The sensing mechanism may correspond to the one described earlier in this document for electrical H_2_ sensors working with Pd NPs on metal oxide: the free electrons injected in WO_3_ result in an increase of electrical conductivity.

Lee et al. [111] developed a H_2_ sensor which gave an eye-readable response at 1% hydrogen in air. The reversible visible coloration change resulted from an interaction of a light beam (300–1100 nm) with the sensing material which places this device at the interface of optical and gasochromic hydrogen sensors. The sensing sample consisted on amorphous WO_3_ with a porous nanocolumnar-like structure deposited on a glass substrate and decorated with Pd NPS. To ensure durability and reproducibility, a passivating PDMS layer was applied on the sample. Interestingly, the authors proposed a method to quantitate the color modification resulting from Equation (12) [111] where L is the lightness value, a is the position on the red-green axis, and b the position on the yellow-blue axis for the initial (1) and final (2) states. The complete change in coloration took around 10 min and the corresponding ΔE value was 52:(12)$$\Delta E_{\mathrm{ab}}=\sqrt{{\left(L^{2}-L^{1}\right)}^{2}+{\left(a^{2}-a^{1}\right)}^{2}+{\left(b^{2}-b^{1}\right)}^{2}}$$

The team of Kalanur et al. [109] also developed a gasochromic hydrogen sensor which showed an irreversible coloration change from white to dark blue upon H_2_ exposure as presented in Figure 21. The irreversible change in color makes the device not reusable but interesting for applications such as inks, paints or pigments that can be printed on paper or polymer substrate for gasochromic applications when mixed with suitable solvents. With the implemented system, they could detect H_2_ concentrations as low as 0.1% in air at room temperature. At this low gas concentration, the maximum coloration change was observed after 10 min. The sensing sample consisted on Pd NPs-decorated MoO_3_ nanoplates deposited on filter paper and the sensing mechanism should be similar to the one reported in [110] and also discussed in the present document. The change in coloration may then be due to the change in oxidation state of Mo from +6 to +5. Moreover, interaction of H atoms with oxygen in the MoO_3_ may result in the theoretical formation of OH_2_ groups in the H_x_MoO_3_ structure [112] and since the formed OH_2_ groups remain in the structure and cannot escape from the lattice, the color remains unchanged and irreversible.

#### 2.2.4. Surface Acoustic Wave (SAW) Sensors

To the best of our knowledge, very few authors have investigated the use of Pd NPs in SAW sensing of hydrogen. Yang et al. [113] reported the use of Pd NPs dispersed on SnO_2_ film supported by a 128° YX LiNbO_3_ piezoelectric substrate. At 175 °C, a response time of about 1 s was found at a hydrogen concentration of 0.2% in N_2_. The SAW sensor was implemented on a test cavity and coaxial-cable was used to connect the SAW device with the radio-frequency unit outside the test chamber to form a closed-loop oscillator. One coupling device was put into the closed-loop oscillator circuit to transmit oscillator signal to the Agilent 53181A frequency counter. To measure the magnitude-frequency and phase frequency characteristics and the insertion loss of SAW device, the sensor was directly connected to network analyzer. The sensor response was evaluated by frequency shift Δf = f_H_ − f_0_, where f_0_ and f_H_ are the center frequency of SAW sensor in dry air and in different concentrations of hydrogen respectively. As reported by Raj et al., the major effects which induce the frequency shift of SAW gas sensors are mass loading effect, elastic loading effect and acoustoelectric loading effect [114]. In the work of Yang et al. [113], the acoustoelectric coupling effect, through changes in the device conductivity in the presence of H_2_, acts as the dominant response mechanism for hydrogen detection. Indeed, prior to H_2_ exposure, oxygen is adsorbed and dissociated by Pd NPs before diffusing into SnO_2_ surface from where it would capture free electrons resulting in ion species. Upon exposure to hydrogen, the hydrogen molecules are dissociated and adsorbed by Pd NPs. Due to a spillover effect, they diffuse into SnO_2_ surface where they interact with adsorbed oxygen ions forming water and releasing electrons to SnO_2_. Thus, the device conductivity increases and therefore slows down the velocity of SAW wave as well as reduces the center frequency of SAW gas sensor [113,115].

Sil et al. [116] also reported the development of a SAW sensor for detection of hydrogen with performances comparable to those reported by Yang et al. [113].

### 2.3. Other Pd Nanostructures for H_2_ Detection

The development of hydrogen detection systems by means of palladium-based nanoparticles have been addressed so far in the present work. It should, however, be pointed out that many other Pd-based nanostructures have been investigated for applications in hydrogen sensing. Particularly, very interesting performances have been reported for the use of palladium-based nanowires (NWs) [117,118,119,120,121] and thin films [122,123,124,125,126].

In 2001, Favier et al. [117] reported the development of a hydrogen sensor working by virtue of the electrical properties of Pd nanowires arrays on cyanoacrylate film. At 5% hydrogen, they achieved to detect the targeted gas with a sensor response time of less than 0.08 s at room temperature and using nitrogen as carrier gas. The sensor response was given by the measured current between two silver contacts spanned by the Pd NWs array when a potential is applied to the silver contacts. The sensor working principle is based on the decrease of resistance in the presence of hydrogen due to the closing of the nanoscopic gaps between the small grains forming the wires. These nanograins are irreversibly formed after a first exposure to H_2_ and subsequent exposure to air. Despite this very good performance regarding the response time, the developed sensor could not detect hydrogen concentration lower than 0.5%.

Yang et al. [118], also working with Pd nanowires for H_2_ detection, developed a sensor with a limit of detection as low as less than 10 ppm. The variations in resistivity upon hydrogenation of devices composed of a single Pd nanowire deposited on glass substrates gave rise to the sensor response which was optimized by the dimensions of the Pd NWs.

A more complex electrical sensor using Pd nanowires was developed by Fang et al. [119]. In their device, Pd NPs were used to interconnect Pd NWs grown in the nanosized channels of nanoporous alumina membrane. Working in air as carrier gas, they achieved to obtain very low response times, typically 1 to 2 s at 0.5% H_2_ and less than 1 s at 1% H_2_. The sensor measurements were performed at room temperature and the sensor presented good response at hydrogen concentration as low as 100 ppm (0.01%). Figure 22 [119] gives a schematic representation of the studied system and the sensor response at low hydrogen concentration. The detection signal was evaluated by the change in system resistance upon hydrogenation. The key element of the obtained high performances is the interconnections between NWs tops that have integrated all of them into a single circuit when they do not touch each other by the side surface.

It is important to mention that the use of bimetallic NWs structures could also help reaching very interesting performances. For example, Gu et al. [120] demonstrated the suppression of hysteresis effect during H_2_ absorption-desorption cycles by using single-crystal PdAu alloy nanowires for optical hydrogen sensing. Similar effects in the use of Pd-Au alloy NPs are confirmed by Wadell et al. [11] and were also discussed previously in the present document. Moreover, alloying gold to palladium allowed, in Gu et al.’s work, tailoring the response and recovery times achieving thus to obtain only 0.5 s and 2 s of respectively response and recovery times when working in the range of 0 to 6.5 % hydrogen concentration. A continuous-wave monochromatic laser with a wavelength of 980 nm was used as detection light and the transmission change upon hydrogenation gave rise to the sensor response.

Li et al. [121] reported the use of platinum-coated palladium single nanowire for electrical detection of hydrogen. They claimed that with an optimized thickness of Pt on Pd NWs, they achieved to detect 0.4% of H_2_ in only 2 s. The sensor limit of detection was as low as 500 ppm and experiments were carried out in air at a range of temperature between 294 and 376 K with better performances at higher temperatures.

A lot of work has also been done in the field of Pd thin film for hydrogen detection. Implementations of several configurations of these films have been investigated for decades. In 1975, Lundström et al. [127] developed a Pd-gate metal-oxide semiconductor (MOS) transistor that was sensitive to H_2_ concentrations as low as 10 ppm in air. Here, a Pd thin film was used as the gate metal and silicon dioxide as the gate oxide on p-type silicon. Hydrogen detection was investigated at 150 °C and it was observed that the sensitivity of the device, that already showed good performances in air, was enhanced when replacing air by argon or nitrogen. The detection relied on the variation of the threshold voltage in presence of hydrogen in the studied atmosphere. The authors explained this variation by the formation of a dipole layer at the metal-oxide interface due to the absorption and diffusion of hydrogen in the Pd film until H atoms reach the mentioned interface. The dipole layer therefore induces changes in the work-function difference between the Pd and SiO_2_ layers and as a consequence the threshold voltage of the MOS transistor is modified [127]. As reported by Sharma [128], this sensing mechanism is common for Pd-gate MOS transistor used for hydrogen detection. Moreover, Stiblert at al. [129] mentioned the same working principle when they developed a Pd-gate MOS transistor for H_2_ leak sensing that could detect very low H_2_ concentration (1 ppm in air) with very small response times (response time of 1 s at 20 ppm H_2_ in air).

Recently, He et al. [122] reported the deposition of Pd nanolayers on butterfly wing scales photonic nanostructure edges for optical hydrogen gas sensing. Experiments were performed at room temperature and using N_2_ as carrier gas. In behalf of the synergetic effect of Pd nanostrips and bio-photonic structures, the Pd-modified butterfly scales allowed achieving an H_2_ limit of detection of less than 10 ppm. Indeed, resulting from the coupling between the plasmonic mode of Pd nanostrips and the optical resonant mode of the wing scale photonic structures, a sharp reflectance peak in the spectra of the Pd-modified butterfly scales was generated and light–matter interaction in Pd nanostructures was enhanced. Upon hydrogen exposure, the gas molecules were adsorbed by the Pd coatings, dissociated into H atoms and absorbed to further form Pd hydride. Following these steps, lattice expansion occurs as well as the change in dielectric properties of Pd nanostrips to less metallic. Thus, the plasmonic absorption of the Pd layers is modified and changes in the reflectance spectral response of the sensing structures are generated [122].

Sanger et al. [126] reported the use of Pd/V_2_O_5_ thin films deposited on glass substrates for electrical hydrogen detection. They achieved detection of H_2_ concentrations in air as low as 2 ppm at 100 °C. At 100 ppm, they obtained a response time of 6 s. The sensing mechanism was similar to the one previously describes in this document for Pd NPs on metal oxide structures. Briefly, before exposure to H_2_, oxygen molecules are pre-adsorbed, dissociated and ionized due to electrons from V_2_O_5_. In the presence of hydrogen, H_2_ gas molecules are similarly adsorbed and dissociated by Pd species and due to a spillover effect, they diffuse onto the surface of V_2_O_5_ where they react with pre-adsorbed oxygen ions to form water vapor molecules, vanadium bronze and releasing electrons to V_2_O_5_. Hence, the sensor resistance is reduced upon hydrogenation [126].

Another configuration of the use of Pd thin films for H_2_ detection is the metal- insulator- metal (MIM) structure consisting in a subsequent stack of metal, insulator and metal layers. Perroton et al. [108] developed a system where a MIM (Au/SiO_2_/Pd) structure is deposited on the core of a multimode fiber and optical hydrogen detection is performed thanks to the spectral modulation of the light transmitted by the fiber. With this configuration, high performances were obtained: response time of 3 s at 4% hydrogen in argon. Downes et al. [123] have also reported similar work where they studied the modelling of optical H_2_ sensing through an Ag/SiO_2_/PdY MIM structure deposited on the core of a multimode optical fiber. They claimed that the use of PdY alloy layer, instead of Pd layer as in Perrotton’s work, improved the sensor durability due to the Y content of the alloy. In these two works, the multilayer thickness defines the sensor performances. The thickness of the first metal (Au in [108] and Ag in [123]) affects the spectral location and intensity of the SPR peak. The SiO_2_ layer modulates the sensor sensitivity and tunes the resonant wavelength. The third layer (Pd in [108] and PdY in [123]) is the hydrogen sensitive layer.

The stack of Pd (or Pd alloy) and other metal layers have also been a research focus of several groups for hydrogen detection applications. For example, Hernàndez et al. [124] reported the use of a stack of Pd and Au nano-layers on a hetero-core optical fiber for optical H_2_ sensing. Upon hydrogenation, the palladium refractive index decreases inducing attenuation changes in the optical fiber evanescent wave and thus in the fiber transmitted signal. Optimization of each layer thickness allowed obtaining a response time of about 5 s for 4% H_2_ concentration. High performances in electrical sensing of H_2_ have been reported by Gautam et al. [125] on account of the use of Pd/Mg/Pd tri-layers, Pd/Mg/Pd/Mg/Pd multi-layers and Pd/Mg-Pd alloy deposited on Si substrates in terms of sensor response time. For these three systems respectively, response times of 4.5, 3.5 and 3 s were obtained at room temperature for hydrogen partial pressure of 2 bar. Mg also being itself known for its sensitivity to hydrogen, taking advantage of the synergetic effect due to the use of both palladium and magnesium- based materials may be very promising for hydrogen sensing applications [125,130].

A lot of research works in the literature addresses the use of Pd-based nanostructures for detection of hydrogen. It will be hard to discuss all of them in the present document. However, Table 5 summarizes the performances of several of them (except for Pd-based nanoparticles which have been widely investigated in this paper). 

Only devices with either a response time of ≤10 s or a limit of detection ≤100 ppm have been selected in this table. It can be seen that apart from Pd-based nanoparticles, not only Pd-based nanowires and thin films have been used for H_2_ sensing applications. For example, Yu et al. [131] developed sensors working with Pd nanotubes and which were able to detect H_2_ concentrations as low as 500 ppm and gave a response time of few seconds at 1% H_2_.

### 2.4. Influence of Specific Parameters on Pd-NPs-based H_2_ Detection

#### 2.4.1. Effects of NPs Density

The density of Pd NPs used for detection of hydrogen is a critical parameter. Several teams [9,46,54,66,72] have investigated the effects of various Pd NPs densities on the H_2_ sensing performances of their devices. In most cases, the control of NPs density is determined by the synthesis parameters. It is usually observed that higher NPs density leads to an enhancement of the sensor response [9,46,66,72] as well as a decrease in the response time [66]. Figure 23 from the work of Han et al. [9] shows three sensor responses associated with three NPs densities and thus illustrates the discussed phenomenon.

The improvement in sensor response with increase in NPs density comes from the large surface area available for gas adsorption when using high NPs density samples [9]. However, a very high density of Pd NPs could induce the formation of a thin film-like structure [46,72]. As highlighted by Han et al. [9], the formation of a Pd thin film should induce a lower surface area available for gas adsorption compared to individual particles since in the last-mentioned case, every particle participates in the gas sensing reaction. The sensor performances after the formation of the Pd thin film may then be reduced. Moreover, in the case of electrical hydrogen sensor working with Pd NPs deposited on substrates which aimed to serve as electronic conduction pathway (carbon material, metal oxides, etc.), the formation of a Pd thin film may induce an electronic conduction pathway located in the Pd thin film instead of the used substrate. This could alter the sensor performances [46,72]. Thus, an optimized NPs density optimizes the sensor performances. Sta et al. [54] reported that the effect of Pd NPs density on the hydrogen sensing performance could also be tailored by modifying the working temperature.

#### 2.4.2. Effects of NPs Size and Shapes

The size of the Pd NPs used for hydrogen detection often plays a key role in the sensing performances of the device. Indeed, due to larger surface area to volume ratio as the particle size decreases, authors usually report higher sensor response for smaller particles when comparing the use of various Pd NPs sizes for detection of hydrogen [7,69] Several authors also reported the decrease of the sensor response time when reducing the size of the Pd NPs [11,69]. Moreover, Seo et al. [69] reported that it is possible to tailor the detection range of the sensor by monitoring the size of the Pd NPs. They achieved to detect very low H_2_ concentration (typically, from 10 to 1000 ppm) with particles of 3 to 5 nm size but the particles became saturated at about 1000 ppm because of their small volume. The use of 10 to 15 nm Pd size particles allowed extending the detection range up to 5% H_2_. This works highlights the strength of Pd NPs size engineering for hydrogen detection. However, as mentioned by Wadell et al. [11], particular attention must be paid when reducing particle size because it inevitably compromises sensor accuracy as signal-to-noise ratio in the sensor readout would grow. Examples of palladium-based nanoparticles, with different sizes and shapes, and that have been studied for H_2_ detection applications are shown in Figure 24.

#### 2.4.3. Effects of Other Gases in the Environment

Selectivity of a sensor is a crucial parameter because it ensures the unambiguous response of the device in an ambient environment where several gases are present. Pd is known for its high sensitivity to hydrogen and to interact specifically with this gas inducing good selectivity of Pd-based sensors for hydrogen detection. This is often verified by performing sensing measurements with the developed devices in presence of O_2_, NH_3_, CH_4_, C_6_H_6_, CO, NO_2_, etc. [6,12,51,52,133,134].

It must be stressed that some parameters may affect the selectivity properties of Pd NPs-based hydrogen sensors: As reported by Gupta et al. [6] and Kabcum et al. [52], the working temperature may alter the selectivity properties of the Pd NPs-based H_2_ sensors. This later team demonstrated that along with an optimal working temperature, an optimal Pd NPs density should be found in order to maximize the selectivity of the device.

To better improve the selectivity of Pd in the development of Pd-based H_2_ sensors, some research teams proposed the use of a coated protective layer that would have a high permeability to hydrogen and a very low one to other gases. The use of a film of metal-organic framework as a protective O_2_ impermeable membrane on Pd nanowires for hydrogen detection has been proposed by Koo et al. [135].

Recently, the team of Chen [134] developed a high-performance hydrogen sensor where a thin layer of PMMA was spin coated onto a Pd NPs film to insure high gas selectivity due to the filtration effect of the PMMA membrane layer. Figure 25 presents a schematic of the sensing sample as well as the selectivity performances of the device in presence of H_2_, CO, CH_4_, and a mixture of H_2_, CO and CH_4_. High sensor signal with a hydrogen selective response was obtained in the presence of the PMMA protective layer even down to 50 ppm H_2_ concentration proving the efficiency and non-disturbing effect of the PMMA film. However, there is a need of optimizing the protective layer thickness since a very thin thickness could result in more permeability to other gases while a thick PMMA layer could drastically lower the response time of the sensor. 

## 3. Conclusion

It is demonstrated in the present work that Pd-based nanostructures have been widely investigated for applications in hydrogen leak detection. Numerous sensing systems take advantage of the change in properties of palladium nanomaterials upon hydrogen exposure to develop such devices. Among them, the literature mostly provides papers addressing the electrical detection of H_2_ using these materials. Nevertheless, optical sensors are also widely studied.

Several configurations of the sensing samples have been selected by the different research teams. In the case of electrical H_2_ sensors, the common ones are the deposition of Pd nanostructures on Si-based substrates, metal oxide substrates or carbon materials. While in the two latter cases, the detection mechanism relies on the fact that the substrate ensures the electron conduction path and the role of the metallic particles is to act as a catalyst in adsorption and dissociation of gas molecules; in the first one, Pd NPs ensures an electron conductive path by closing the gaps between particles thanks to volume expansion upon H_2_ exposure.

For Pd NPs-based optical H_2_ sensors, the most encountered configurations are the deposition of the NPs on Si-based substrates or on the core of an optical fiber. The sensor response arises from the variation of system optical responses upon hydrogenation when an electromagnetic beam is sent directly on the metallic particles (in the case of Pd on Si substrates) or interacts with the palladium through an evanescent beam (in the case of Pd on optical fiber).

In term of response time and limit of detection, excellent performances, such as limit of detection lower than 5 ppm H_2_ concentration and less than one second for response time, have been reported in the investigation of Pd NPs electrical H_2_ sensors. However, the use of electrical sensors comes along with the risk of unguaranteed safety and sensor longevity issues due to the use of electrical contact in harsh environment. The use of optical sensors could help overcome these problems.

It should be mentioned that until now, Pd NPs-based optical H_2_ sensors hardly achieve the excellent performances (LOD < 10 ppm; response time < 1 s) reported for electrical devices. A window of optimization is still widely opened for these optical sensors. One way of improving their performances could be to mimic, as nanoparticles, the structures of Pd-based film materials already developed for H_2_ optical sensing. Due to the increased surface to volume ratio, the use of Pd NPs may provide better performances resulting from higher effective surface available for interaction of Pd with the hydrogen gas molecules. It has also been demonstrated that the use of bimetallic structures could help optimize the performances.

## Figures and Tables

**Figure 1 sensors-19-04478-f001:**
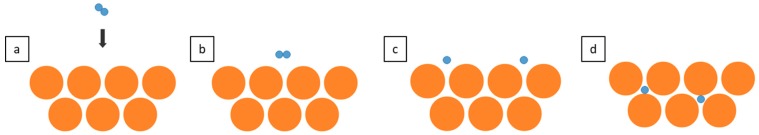
Reaction steps between hydrogen gas molecules and Pd samples. (**a**) H_2_ molecule approaching the metal surface. (**b**) Interaction of the H_2_ molecule by Van der Waals forces (physisorbed state). (**c**) Chemisorbed hydrogen after dissociation. (**d**) Occupation of subsurface sites and diffusion into bulk lattice sites. Adapted from [19].

**Figure 2 sensors-19-04478-f002:**
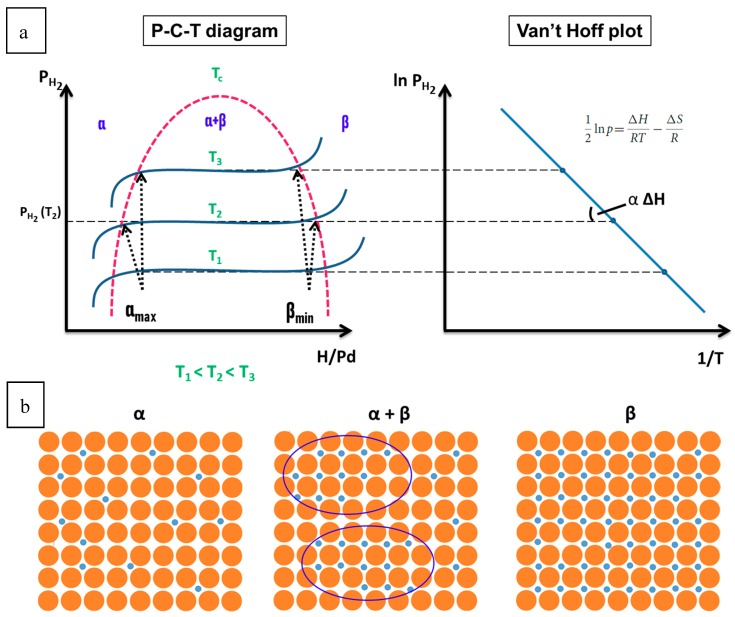
(**a**). Schematic representation of an ideal Pd P-C-T diagram and the corresponding Van’t Hoff plot. (**b**). Illustration of the phases formed during hydrogenation of Pd samples. Grey spheres represent Pd atoms while red ones illustrate H atoms. Adapted from [19].

**Figure 3 sensors-19-04478-f003:**
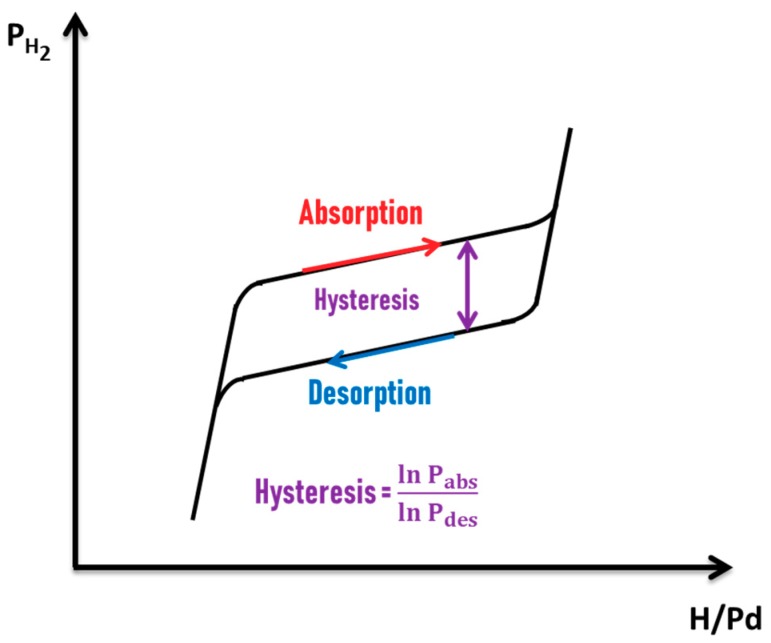
Schematic representation of a real isothermal measurement of a Pd P-C-T diagram. Adapted from [18].

**Figure 4 sensors-19-04478-f004:**
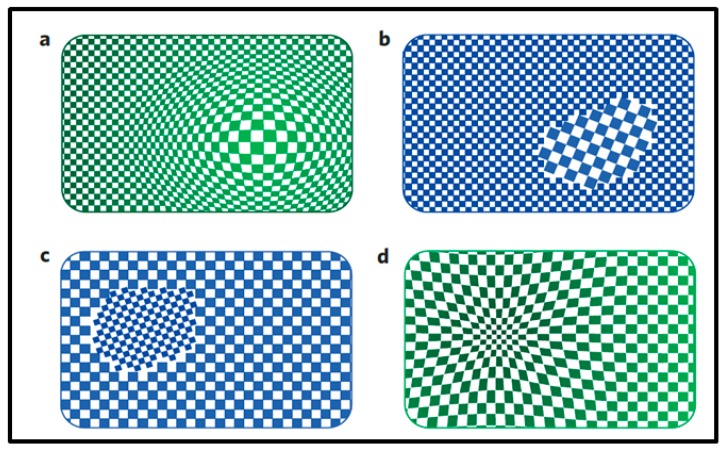
(**a**). Schematic coherent α to β phase transformation characterized by a smooth variation of lattice parameter in the single metastable state. There is no coexistence of the α and β PdH*x* phases. (**b**). Schematic incoherent α to β phase transformation characterized by nucleation of β PdH*x* phase in the dilute α PdH*x* phase accompanied with formation of dislocation in the crystalline structure. The two phases coexist. (**c**). Schematic incoherent β to α phase transformation. (**d**). Schematic coherent β to α phase transformation. Reprinted with permission from [27]. Copyright 2016 Nature Materials.

**Figure 5 sensors-19-04478-f005:**
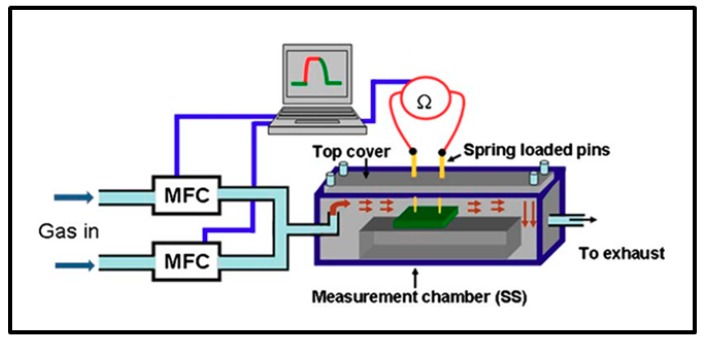
Typical H_2_ electrical sensing experiment setup. The green component in the measurement chamber is the sensing sample. Reprinted with permission from [7]. Copyright 2009 Nanoscale Research Letters.

**Figure 6 sensors-19-04478-f006:**
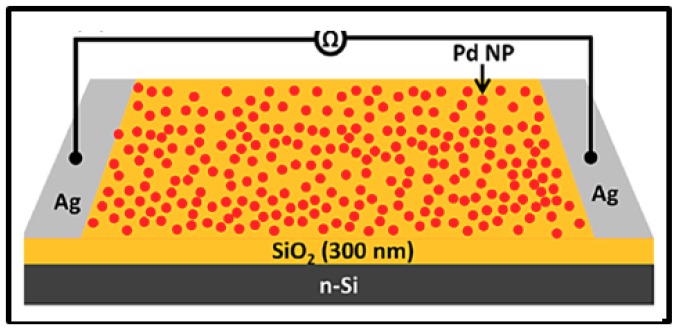
Typical H_2_ sensing system with the configuration of Pd NPs on SiO_2_. Reprinted with permission from [39]. Copyright 2015 Sensors and Actuators B: Chemical.

**Figure 7 sensors-19-04478-f007:**
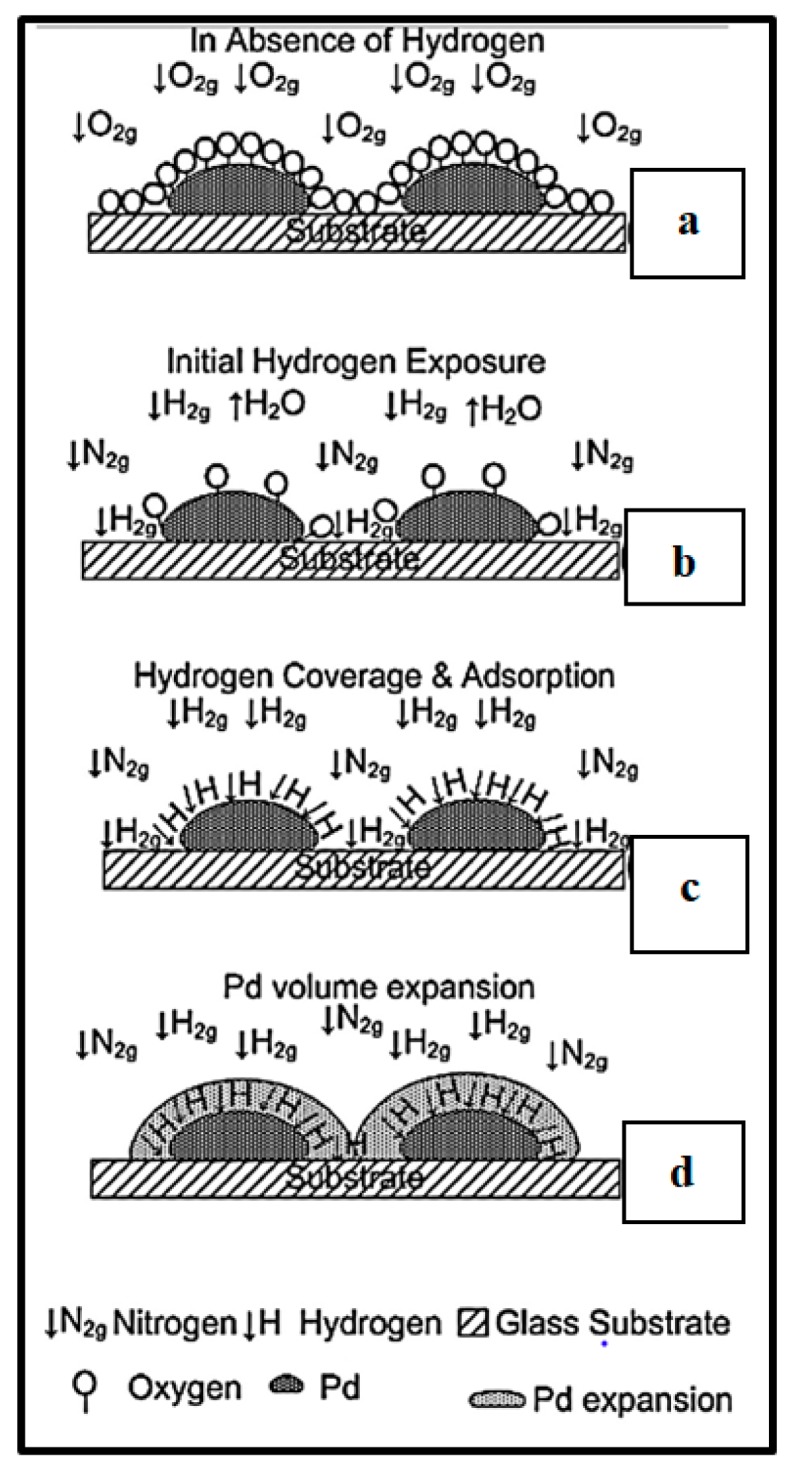
Hydrogen sensing mechanism when using Pd NPs on Si-based substrates as sensing material: (**a**) In absence of hydrogen, the surface and the interface between two Pd NPs are covered by a layer of spilled over charged oxygen species; (**b**) During initial hydrogen exposure, H_2_ molecules interact with oxygen species resulting in water formation. Therefore, oxygen species are removed from the surface and vacant adsorption sites are formed; (**c**) The vacant surface sites are then gradually filled by hydrogen that will diffuse in the Pd matrix; (**d**) Formation of palladium hydride leads to a volume expansion of Pd NPs that reduces the gap between two particles until they touch each other. Reprinted with permission from [6]. Copyright 2014 Sensors and Actuators B: Chemical.

**Figure 8 sensors-19-04478-f008:**
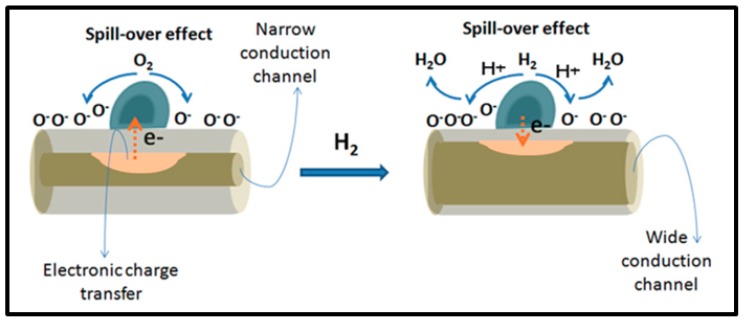
Illustration of hydrogen sensing mechanism of Pd NPs on MO_x_ sensor. First stage, oxygen is adsorbed and dissociated by Pd NPs. Due to a spillover effect, it diffuses onto the MO_x_ surface from where it would capture free electrons resulting in ion species and formation (or enlargement) of a depletion layer. Upon hydrogen exposure, H_2_ gas molecules are similarly adsorbed and dissociated by the metal and they diffuse onto the surface of the MO_x_ where reaction with pre-adsorbed oxygen ions gives water vapor molecules and electrons that are released, reducing the depletion layer and increasing the device conductance. Reprinted with permission from [51]. Copyright 2016 Thin Solid Films.

**Figure 9 sensors-19-04478-f009:**
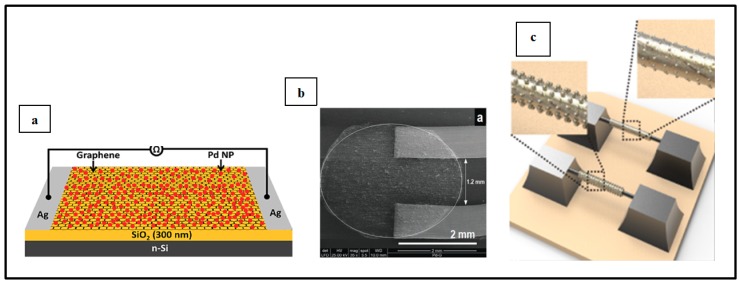
H_2_ sensing systems with the configuration of Pd NPs on carbon material. (**a**) Schematic diagrams of Pd NPs/graphene-based sensors. Reprinted with permission from [39]. Copyright 2015 Sensors and Actuators B: Chemical. (**b**) As-fabricated H_2_ sensors: Pd NPs/GO thin films on a glass substrate between two Cu contacts. Reprinted with permission from [70]. Copyright 2015 New J Chem. (**c**) Schematic of single suspended PdNPs/carbon nanowires integrated onto a H_2_ detection chip. Reprinted with permission from [69]. Copyright 2017 Sensors and Actuators B: Chemical.

**Figure 10 sensors-19-04478-f010:**
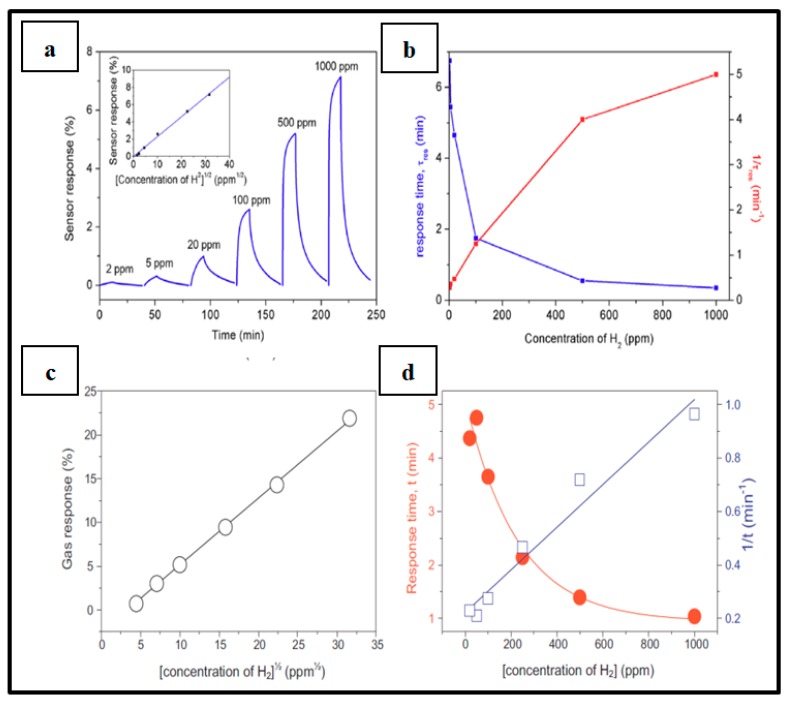
(**a**,**b**) H_2_ sensor based on Pd NPs on CNT. Reprinted with permission from [68]. Copyright 2018 International Journal of Hydrogen Energy. (**a**). H_2_ sensing response for different H_2_ concentrations. Inset shows that sensor response was proportional to the square root of H_2_ concentration. (**b**) Evolution of response time with H_2_ concentration. (**c**,**d**) H_2_ sensor based on Pd NPs on graphene. Reprinted with permission from [46]. Copyright 2012 Sensors and Actuators B: Chemical. (**c**) H_2_ sensing response as a function of square root of H_2_ concentration. (**d**). Evolution of response time with H_2_ concentration.

**Figure 11 sensors-19-04478-f011:**
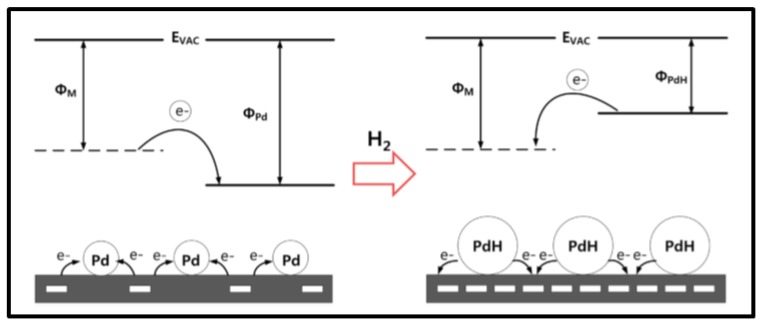
Energy band diagrams of Pd and MoS_2_ before and upon H_2_ exposure. Lower work function of MoS_2_ in comparison to Pd results in electron transfer to Pd before H_2_ exposure. Upon H_2_ exposure, lower work function of PdH_x_ in comparison to MoS_2_ results in electron transfer to MoS_2_ and hence, compensation of charge carriers is observed in MoS_2_. Reprinted with permission from [72]. Copyright 2017 Sensors and Actuators B: Chemical.

**Figure 12 sensors-19-04478-f012:**
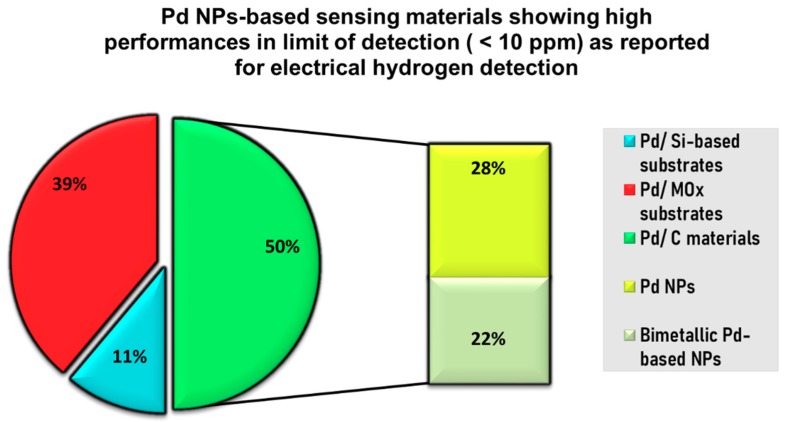
Schematic of proportions of Pd NPs-based hydrogen sensing materials that show high limit of detection performances as reported in the literature for electrical H_2_ detection.

**Figure 13 sensors-19-04478-f013:**
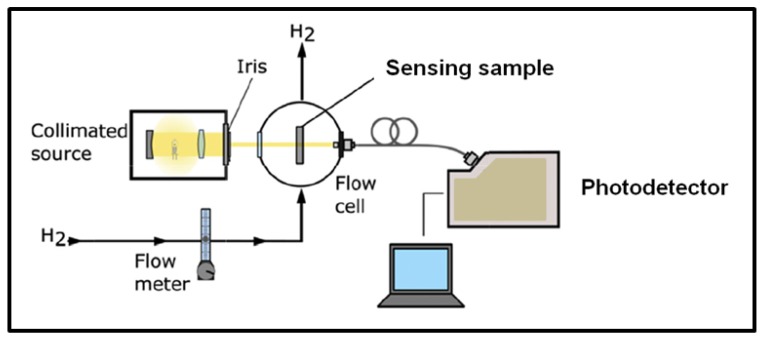
Typical H_2_ optical sensing experiment setup. Reprinted with permission from [15]. Copyright 2018 International Journal of Hydrogen Energy.

**Figure 14 sensors-19-04478-f014:**
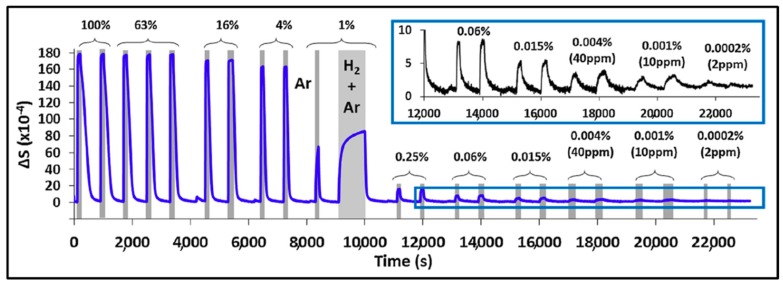
Watkins et al. ‘s sensor signal measured at 885 nm during alternative cycles between pure Ar and different decreasing concentrations of H_2_ in Ar. Reprinted with permission from [12]. Copyright 2018 Sensors and Actuators B: Chemical.

**Figure 15 sensors-19-04478-f015:**
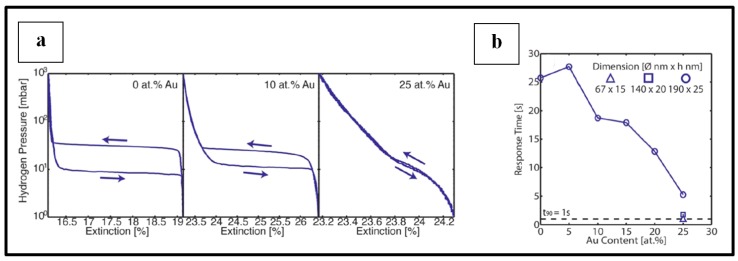
Results from C. Wadell et al.’s work. (**a**) Hydrogen absorption and desorption isotherms for three different alloy compositions. (**b**) Reduction of the response time as the Au content increase and also by reducing the NPs size. Reprinted with permission from [11]. Copyright 2015 Nano Letters.

**Figure 16 sensors-19-04478-f016:**
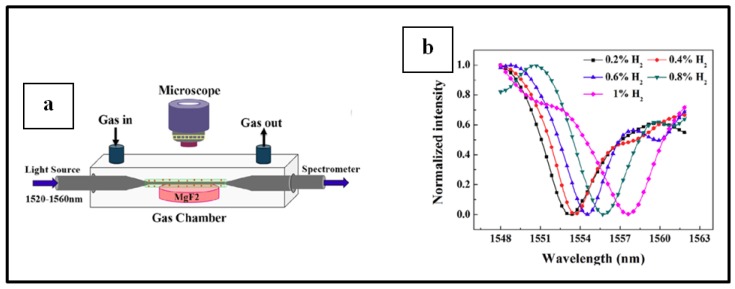
(**a**) Experimental set up used by Li et al. (**b**) Spectra change for a microfiber hydrogen sensor when H_2_ concentrations varied from 0.2 to 1 vol%. Reprinted with permission from [17]. Copyright 2018 Materials Letters.

**Figure 17 sensors-19-04478-f017:**
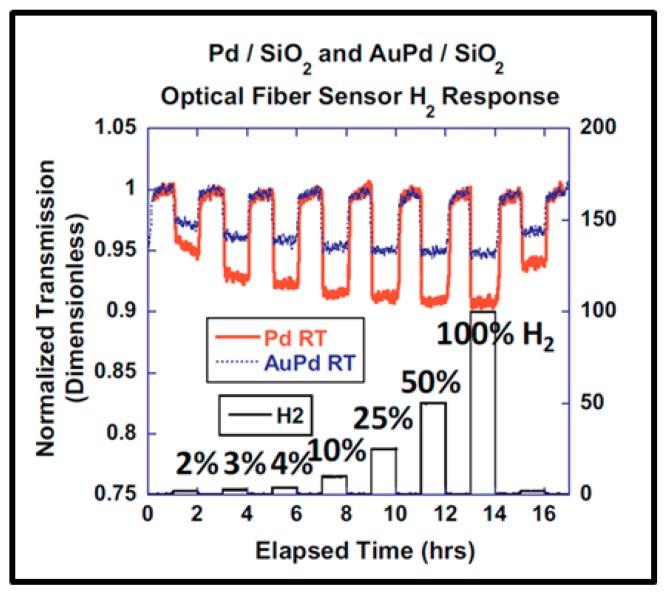
Corresponding sensing responses measured for Pd/SiO_2_ and AuPd/SiO_2_ optical fiber sensors at room temperature. Reprinted with permission from [89]. Copyright 2015 Sensors and Actuators B: Chemical.

**Figure 18 sensors-19-04478-f018:**
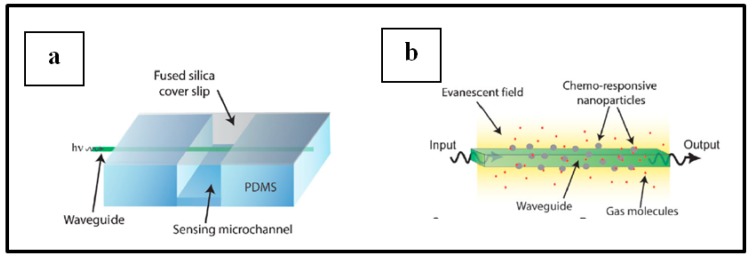
(**a**) Schematic of sensing setup. (**b**) Zoom in of the evanescent field sensing region of the waveguide. Light is coupled into the SnO_2_ cavity and travels through the sensing region which is loaded with chemo-responsive nanoparticles. Reprinted with permission from [105]. Copyright 2008 Advanced Materials.

**Figure 19 sensors-19-04478-f019:**
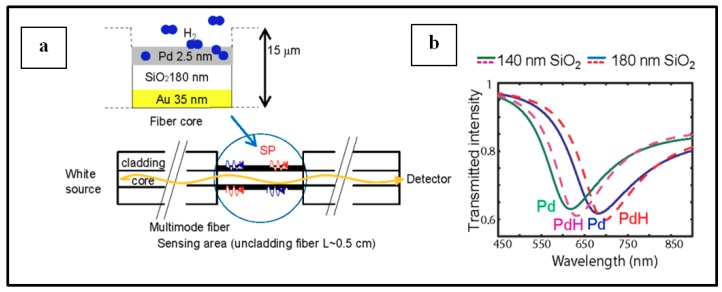
(**a**) Schematic representation of the way the MIM structure on the fiber core, after removing the cladding. (**b**) The simulated transmitted intensity as a function of the wavelength for two different SiO_2_ thicknesses. The line and the dashed line represent respectively the metallic and the hydrogenated states. Reprinted with permission from [108] © The Optical Society

**Figure 20 sensors-19-04478-f020:**
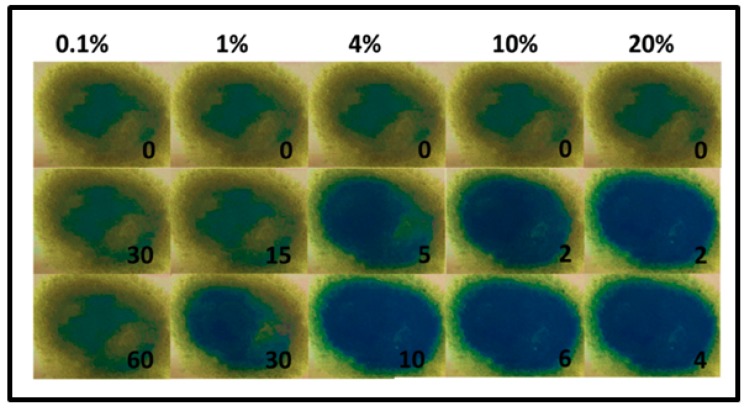
Photographic images of 2D WO_3_/Pd on filter paper during gasochromic H_2_ detection tests with different concentrations of H_2_ at different time intervals. Response times in seconds are indicated in respective photographic image. Reprinted with permission from [110]. Copyright 2017 International Journal of Hydrogen Energy.

**Figure 21 sensors-19-04478-f021:**
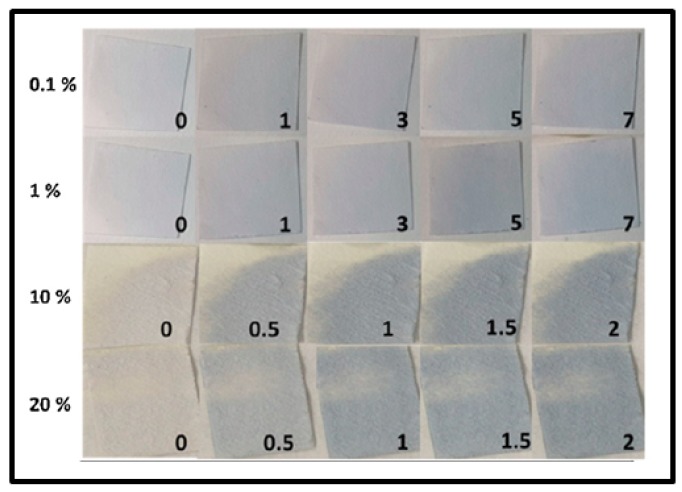
Photographic images of MoO_3_-Pd on filter paper during gasochromic H_2_ detection tests with different concentrations of H_2_ at different time intervals at room temperature (25 °C). Time in min at which the picture was taken is indicated in the down right corner. Reprinted with permission from [109]. Copyright 2017 Sensors and Actuators B: Chemical.

**Figure 22 sensors-19-04478-f022:**
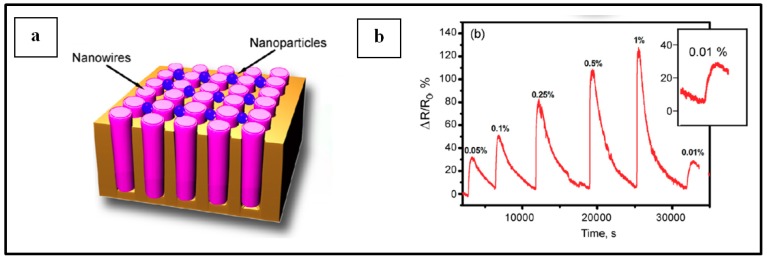
(**a**) Schematic representation of the H_2_ sensing system developed by Fang et al. (**b**) Sensor response to different percentage of hydrogen in air flow. Reprinted with permission from [119]. Copyright 2015 International Journal of Hydrogen Energy.

**Figure 23 sensors-19-04478-f023:**
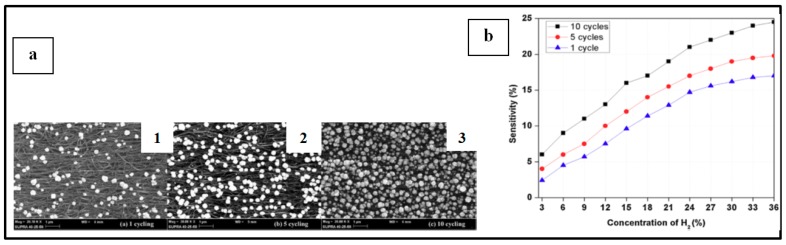
(**a**) SEM images of Pd NPs deposited on MWCNTs by electrodeposition for (1) 1, (2) 5, and (3) 10 cycles. Pd NPs density increases with the number of electrodeposition cycles. (**b**) Sensitivities of MWCNTs/Pd sensors as a function of H_2_ concentration with respect to number of deposition cycles (increasing density). Reprinted with permission from [9]. Copyright 2014 Chemical Physics Letters.

**Figure 24 sensors-19-04478-f024:**
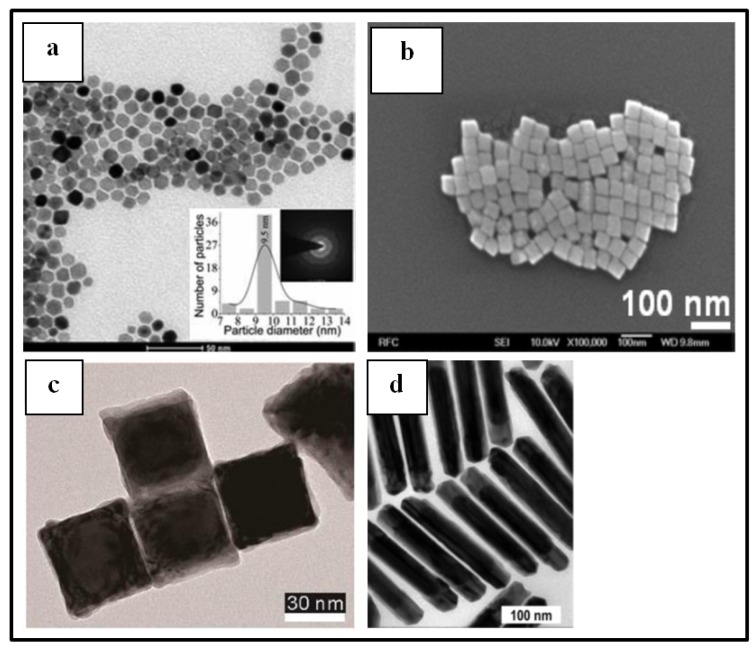
**(a**) TEM image of Pd nanoparticles (~9.5 nm) used to decorate Al-doped ZnO surfaces for H_2_ detection. Reprinted with permission from [58]. Copyright 2015 Journal of Applied Physics. (**b**) From SEM image of Pd nanocubes (~70 nm) used to decorate graphene sheets for H_2_ detection. Reprinted with permission from [67]. Copyright 2014 Sensors and Actuators B: Chemical. (**c**) TEM image of Au@Pd nanocubes (~48 nm) used for optical H_2_ detection. [10]. Copyright 2018 Sensors. (**d**) TEM image of Au@Pd nanorods investigated for optical H_2_ detection. Reprinted with permission from [132]. Copyright 2016 Chemistry of Materials.

**Figure 25 sensors-19-04478-f025:**
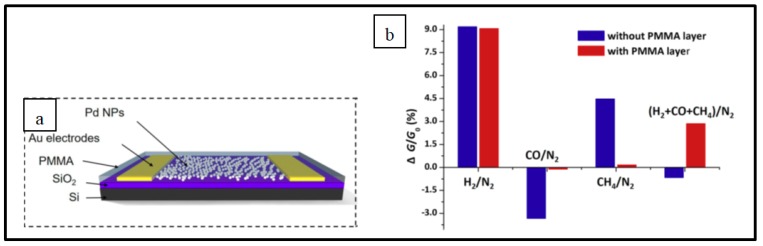
(**a**) Schematic of the sensing sample used in [134]. (**b**) Response of sensors with and without a PMMA membrane layer to target gas mixtures including CO/N_2_, CH_4_/N_2_, H_2_/N_2_, and (H_2_ + CO + CH_4_)/N_2_. Gas concentrations were 1000 ppm. Reprinted with permission from [134]. Copyright 2017 ACS Appl Mater Interfaces.

**Table 1 sensors-19-04478-t001:** DOE Target Specifications for Hydrogen Safety Sensors. Adapted from [13,14].

Parameter	Specification (Value)
Measurement range	0.1 to 10 vol%
Operating temperature	−30 °C to 80 °C
Response time	<1 s
Accuracy	5% of full scale
Gas environment	Ambient air, 10% to 98% relative humidity
Lifetime	10 years
Interference Resistance	(e.g., to hydrocarbons)

**Table 2 sensors-19-04478-t002:** Comparison between H_2_ sensing in Air and in N_2_. Reprinted with permission from [6]. Copyright 2014 Sensors and Actuators B: Chemical.

Temperature	Concentration	Hydrogen in Nitrogen			Hydrogen in Air		
		Response (%)	Response Time (s)	Recovery Time (s)	Response (%)	Response Time (s)	Recovery Time (s)
50 °C	1%	15.4	10	31	10.2	40	430
	0.50%	9.1	8	24	4.39	44	340
	0.40%	7.24	7	24	3.63	44	280
	0.30%	5.11	4	20	2.51	40	234
	0.20%	2.73	3	18	1.2	41	150
	0.10%	1.15	3	10	0.48	33	82

**Table 4 sensors-19-04478-t004:** Summary of best performances of Pd NPs-based optical hydrogen gas sensors found in the literature.

Sensing Material	Limit of Detection	Response Time	Recovery Time	T°	Ref.
Pd NPs/fused silica substrate	nr	2s/5%	5 s	RT	[15]
PdAu NPs/glass substrate	~0.1%	<10 s/5%	<20 s	RT	[95]
Au@Pd NPs/quartz	~0.1%	4 s/4%	30 s	RT	[10]
Pd NPs/glass substrate	<10 ppm	32 s/4%	nr	RT	[12]
PdAu nanodisks/glass substrate	~0.1%	<1 s/0.1–4%	nr	RT	[11]
PdAu NPs/optical microfiber	<1%	2 s/4%	20 s	RT	[100]
Pd NPs - PMMA/optical microfiber	~35.8 ppm	5 s/0.2–1%	nr	RT	[17]
Pd NPs/SnO_2_ waveguide	~0.5%	3 s/3%	2 s	RT	[105]

**Table 5 sensors-19-04478-t005:** Summary of the best performing Pd-based hydrogen gas sensors found in the literature. Pd NPs-based hydrogen sensors are not taken into account in this table.

Sensing Material	Limit of Detection	Response Time	Recovery Time	T°	Sensor Type	Ref.
Pd nanowire	0.50%	<0.08 s/~4%	nr	RT	electrical	[114]
Pd nanowire	27 ppm	4 s/~2.4%	nr	RT	electrical	[126]
Pd nanowire	<10 ppm	~ 30 s/~4%	100 s	RT	electrical	[115]
Pd nanowire	200 ppm	2 s/~5%	6 s	RT	electrical	[127]
Pd nanowire	100 ppm	<1 s/~1%	nr	RT	electrical	[116]
PdAu nanowire	0.20%	0.5 s/0–6.5%	2 s	RT	optical	[117]
Pd@Pt nanowire	500 ppm	2 s/~0.4%	2.5 s	294–376 K	electrical	[118]
Pd nanostrip	<10 ppm	12 s/~1%	nr	RT	optical	[119]
Au/SiO_2_/Pd MIM	0.50%	3 s/~4%	10 s	RT	optical	[105]
Pd/Au films	nr	5 s/4%	13 s	nr	optical	[121]
Pd/Mg film	<10 ppm	6 s/1%	33 s	RT	electrical	[124]
Pd/Mg-Pd film	nr	3 s/2 bar	~3 s	RT	electrical	[122]
Pd/Mg-Ni film	10 ppm	5 s/0.1%	nr	RT	electrical	[128]
Pd-V_2_O_5_ film	2 ppm	6 s/~100 ppm	21 s	373 K	electrical	[123]
Pd/ZnO film	0.10%	0.3 s/2%	18 s	353 K	nr	[51]
Pd nanotube	500 ppm	few s/1%	nr	RT	electrical	[125]
PdY film	500 ppm	4 s/~4%	nr	RT	optical	[129]
Pd gratings	50 ppm	18 s/~0.35%	nr	RT	electrical	[130]
Pd gratings	nr	4 s/~1%	nr	293–323 K	optical	[131]
Pd NWs@ZIF	600 ppm	7 s/~1%	10 s	RT	electrical	[132]
Pd film/SiC nanocauliflowers	2 ppm	7 s/~100 ppm	13 s	573 K	electrical	[133]
Pd film	48 ppm	15 s/~0.2%	nr	RT	SAW	[134]
Pd/ZnO film	59 ppm	12 s/2%	nr	RT	SAW	[112]
PdAu film	100–300 ppm	8 s/1%	2 min	RT	optical	[135]
Pd film (Pd-gate MOS transistor)	10 ppm	nr	nr	423 K	electrical	[127]
Pd film (Pd-gate MOS transistor)	1 ppm	1s/20 ppm	3 s	423 K	electrical	[129]

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
