# Peer review of "Recent Advances in Palladium Nanoparticles-Based Hydrogen Sensors for Leak Detection"

_sensors, 2019, doi:10.3390/s19204478_

Round 1

Reviewer 1 Report

This article is an important work (in quantity and quality) on the recent developments of hydrogen sensors in its most currently studied form,  "Palladium nanoparticles-based". After a presentation on the physico-chemical principles of these sensors, the authors review recent works (after 2012) on different transduction technologies:

Electrical (about 40 references) Optical (about 20 references) Various (gasochromic, SAW etc… about 20 references)

For each type of transduction, a table summarizes the main performances of the sensors.

However, some details need to be reviewed::

The title is surprising because it is limited to a single application, leak detection, whereas few articles specify this application and it is often highlighted as having detection limits of a few tens of ppm... References in the text should be reviewed: number in brackets and especially punctuation after. L118: Pd hybride -> hybrid Pd L215: variations -> variation L277: The definition of the answer presented here (relative variation of resistance, from 0 to 100%) is not always the one used in the articles: some use the ratio of resistance (55…), some the relative variation of conductance (31…). You must always indicate the correct term in the text, and in any case, use "answer" rather than "signal" (488, 489 at least). In Table 2, you should recalculate some "answers" so that they are all compatible with each other, which is not currently the case. It should be mentioned somewhere that the limit of detection (LOD) used here is an order of magnitude of the smallest source of observable variation, and not the rigorously defined metrological characteristic (and calculated as 3 times the standard deviation). Use the same term everywhere, and therefore choose between “detection limit” and “limit of detection”. L661-665: not understandable (repetition?) Fig 15: overlaps text. L1044-1045: table too big without title and number. In the H2 formula, often "2" is not in subscript, especially in the conclusion.

Reviewer 2 Report

General comments:

Good manuscript reviewing Pd-based sensors for hydrogen detection. The manuscript is well written.

Please, review the references indication throughout the text, according the SENSORS rules. For example, in line 28 the correct indication should be “...energy [1,2]. It is...”.

Authors use the words “latter” and “thank” many times. Not wrong, but better to avoid repetition.

Suggestion: To enrich the manuscript, authors may include some review of Hydrogen sensors based on Pd-gate MOS-FET transistors.

Specific comments:

L.32: It is not necessary to inform that Paris is the capital of France.

L.80: Item 2 is not “Results”. It is better change to something like “2 – Review of Palladium-based Hydrogen sensors”

Fig. 2a: I suppose the arrows indicating “alpha-max” and “beta-min” should be pointed for the beginning and the end of the horizontal lines, respectively.

L.194: Please, correct the word “electronic”.

L.446: Please, correct the word “carbon”.

Fig. 9 is a big mess. Please rearrange it. There are repetitions and figures 9d, 9e, and 9f are not mentioned neither in the text nor in the figure caption.

L.1044: There is a lack of the Table 4 title.

Reviewer 3 Report

The article presents a review of literature data on electrical and optical sensors based on palladium nanoparticles for hydrogen detection and also includes the information on the mechanism of sensor response formation in different cases. As the parameters by which the authors compare the sensors, the detection limit and response time are chosen. In order to more rigorously substantiate the numerical values ​​of these characteristics, it is necessary to clarify in the Introduction the tasks (environmental, technological etc.) in which it is necessary to detect hydrogen, and indicate:

(i) What are the requirements concerning the concentration range that needs to be detected?

(ii) In  what atmosphere it is necessary to effectuate the analysis (air, inert atmosphere, humidity level, etc.)?

(iii) What should be the response speed of the sensor in different tasks?

(iv) What are the limitations for the working temperature?

Round 2

Reviewer 3 Report

The article may be accepted in its present form/